# UNITE: Universal kNowledge Integration from Task-specific Experts

**Shuxia Lin, Qiufeng Wang, Xu Yang,**[*] **Xin Geng**[*]
School of Computer Science and Engineering, Southeast University
Nanjing 210096, China
Key Laboratory of New Generation AI Technology and Its Interdisciplinary Applications
Ministry of Education, China
{shuxialin, qfwang, xuyang_palm, xgeng}@seu.edu.cn

## Abstract

Large language models (LLMs) with Mixture-of-Experts (MoE) architectures achieve strong performance under sparse activation. However, their expertise is often fragmented across experts and redundant across layers. Prior studies primarily diagnosed redundancy or parameter importance, revealing overlaps but lacking mechanisms to transform them into reusable knowledge. In contrast, human learning succeeds not by memorizing isolated facts but by reusing shared strategies across domains, which motivates the question: do MoE models similarly encode universal knowledge that can be systematically extracted and reused? We propose Universal kNowledge Integration from Task-specific Experts (*UNITE*), a framework that consolidates experts through Fisher-weighted fusion and then applies Tucker decomposition to disentangle shared low-rank input/output subspaces as universal knowledge from layer-specific variations. This universal component provides a compact basis for reconstructing target models with flexible depth, enabling lightweight yet competitive adaptation across tasks. To assess effectiveness, we evaluate data efficiency, convergence speed, and generalization across multiple MoE-based LLMs and diverse datasets. The results show that *UNITE* not only extracts universal knowledge, but also flexibly enabling once-for-all extraction and flexible target model construction that generalize across domains.

## 1 Introduction

Large language models (LLMs) equipped with Mixture-of-Experts (MoE) architectures have achieved remarkable success across a wide range of tasks (Liu et al., 2024; Zhu et al., 2024a; Wang et al., 2025). In these architectures, inputs are routed to only a small subset of experts, where each expert captures distinct modes of reasoning and problem-solving. This routing is typically implemented using top-1 or top-2 gating, which activates merely one or two experts out of dozens per layer (Fedus et al., 2022; Du et al., 2022), meaning that less than 5–10% of the parameters participate in processing any given token. While this sparse activation brings substantial computational savings and scalability, it also implies that much of the expert knowledge remains unused for most inputs. Consequently, expertise in MoE models is highly fragmented across layers and experts, raising a central question: *beyond their diversity, do these experts also encode task-agnostic knowledge—universal structures that transcend individual experts and tasks?*

This question resonates with the way humans learn, since effective learning is not achieved by memorizing isolated facts but by acquiring transferable strategies, abstractions, and reasoning principles that transcend disciplinary boundaries (Anderson & Krathwohl, 2001; Hilton & Pellegrino, 2012), as illustrated in Figure 1a. For instance, a student who practises solving mathematical problems develops logical deduction and abstraction skills that later support reasoning in physics or computer science. These higher-order abilities illustrate how humans consolidate domain-specific experiences into shared cognitive principles that remain useful across diverse tasks. By analogy, MoE-based LLMs may also exhibit similar properties: although each expert is designed to specialize, their pa-

---

[*]Co-corresponding author.

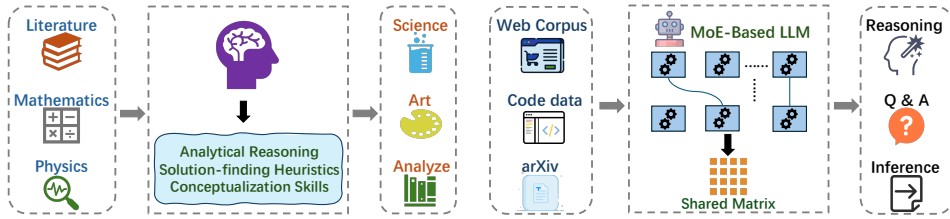

Figure 1: Comparison between human learning and universal-knowledge extraction in MoE-based LLMs. **(a)** Humans integrate domain knowledge into transferable reasoning skills applied across tasks. **(b)** LLMs trained on diverse corpora encode fragmented expertise across experts; the *UNITE* framework extracts shared universal-knowledge to support various downstream tasks.

rameters are not entirely independent and may contain overlaps, redundancies, and recurring transformations that implicitly capture transferable patterns. If such universal knowledge exists, it could provide a compact foundation that can be systematically extracted and reused, echoing the idea of *learngenes* (Wang et al., 2022; 2023) as inheritable structural knowledge units, much as humans reuse shared strategies when facing new problems, as illustrated in Figure 1b.

Previous studies on large models have examined parameter redundancy and importance using techniques such as gradient-based sensitivity analysis (Molchanov et al., 2019; Sanh et al., 2020), loss-change tracking (Frankle & Carbin, 2018), and data attribution methods like Data Shapley (Zhang et al., 2024; Xu et al., 2023). Sensitivity-based approaches estimate parameter importance by monitoring gradient magnitudes or their effect on loss, whereas data valuation methods attribute predictions to training samples to assess which parameters or subsets of data contribute the most to performance. These studies show that large models contain overlapping or compressible structures, indicating the existence of shared information. However, these approaches mainly identify or interpret influential parameters for a specific task or dataset, rather than explicitly extracting generalizable knowledge. Moreover, the importance patterns they reveal are often either highly *dispersed* across individual parameters or aggregated at the *whole-layer* level, which makes it challenging to transform these findings into systematically reusable knowledge across tasks.

To bridge this gap, we propose Universal kNowledge Integration from Task-specific Experts (*UNITE*), a framework designed to systematically explore, extract, and validate universal knowledge within MoE-based LLMs. *UNITE* is motivated by two key observations. First, expertise in MoE models is highly fragmented: since only a small subset of experts is activated per input, large portions of model capacity remain underutilized, and parameters across experts often contain redundancies (Du et al., 2022; Mu & Lin, 2025). This fragmentation makes it difficult to directly identify shared structures, highlighting the need for systematic consolidation. Second, redundancy is not limited to individual layers but also persists across them, as many encode overlapping or functionally similar transformations (Gromov et al., 2024; Csordás et al., 2025). These observations suggest that universal knowledge should be considered both within and across layers, where higher-order generalizations may emerge.

To operationalize these ideas, *UNITE* proceeds in two stages. At the intra-layer level, where expertise in MoE layers is fragmented, we compute Fisher information scores (LeCun et al., 1989; Martens, 2020) on a diverse calibration dataset and fuse experts through Fisher-weighted averaging, yielding $L$ experts in an $L$-layer model. While this reduces fragmentation, many layers still encode overlapping transformations (Gromov et al., 2024; Csordás et al., 2025). To capture higher-order regularities, *UNITE* stacks all fused experts into a tensor and applies Tucker decomposition (Kolda & Bader, 2009; Novikov et al., 2015), which separates shared low-rank input/output subspaces, representing universal bases reused across layers, from layer-specific coefficients that capture local variations. These shared subspaces represent universal knowledge, consistent with the idea that factor matrices encode reusable subspaces (Kolda & Bader, 2009; Aghajanyan et al., 2020) and that Transformers tolerate cross-layer parameter sharing (Dehghani et al., 2018; Lan et al., 2019). Together, these stages transform fragmented expert parameters into compact, reusable universal knowledge, which can be viewed as learngenes.

Most prior explorations have focused on identifying important parameters or diagnosing redundancy, but have paid little attention to how such knowledge can be systematically reused. In contrast, much

like human learning where acquired principles transfer across domains at low cost, we aim not only to extract but also to leverage universal knowledge. After obtaining a compact, shared component through consolidation and decomposition, we treat this component as an inheritable learngene and recompose the feed-forward modules by combining the shared bases with lightweight, layer-specific coefficients, which are randomly initialized and subsequently trained on different downstream datasets. This construction enables target models whose depth can be flexibly scaled to match computational budgets while consistently preserving the same universal component. To evaluate the effectiveness of the extracted knowledge, we adapt three criteria that mirror hallmarks of human learning: data efficiency (achieving strong performance with fewer training samples), convergence speed (requiring fewer optimization steps), and generalization (maintaining accuracy across diverse tasks). These metrics allow us to directly test whether the extracted knowledge provides a reusable foundation for efficient adaptation in MoE-based LLMs.

We validate the effectiveness of *UNITE* through experiments on three representative MoE-based LLMs. For each source model, we construct target models and evaluate the transferability of extracted universal knowledge across datasets in scientific reasoning, commonsense inference, and domain-specific knowledge, under three criteria: data efficiency, convergence speed, and generalization. UNITE-based models consistently outperform randomly initialized baselines, with gains exceeding $+6\%$ on reasoning tasks. They also rival or surpass widely used small- and medium-scale pre-trained models with fewer parameters, and remain competitive against compression methods. Notably, on PIQA the DeepSeek-based UNITE model (317M, 0.629) outperforms both its compressed baseline (990M, 0.625) and BERT-Large (340M, 0.505). Beyond these, *UNITE* supports flexible target model construction at varying depths, and ablation studies verify the importance of its key components. Together, these results show that *UNITE* extracts transferable universal knowledge and leverages it to build scalable, efficient, and competitive target models across diverse scenarios.

## 2 RELATED WORK

**Mixture-of-Experts in LLMs.** With the rapid scaling of large language models (LLMs) (Liu et al., 2025; Peng et al., 2025; Yang et al., 2026; Cao et al., 2024; Wu et al., 2025b; Kou et al., 2025a), modern architectures have evolved toward increasingly over-parameterized and modular designs to improve capacity and efficiency. Mixture-of-Experts (MoE) architectures scale large language models by routing each input token to a small subset of experts per layer, typically via top-$k$ gating with $k = 1$ or $2$ (Du et al., 2022; Wang et al., 2025). This conditional computation provides substantial efficiency gains while preserving accuracy, enabling trillion-parameter models with a feasible inference cost (Vats et al., 2024; Cai et al., 2025; Lu et al., 2025). However, since fewer than 5–10% of experts are active for any token, much of the capacity remains underutilized, leading to fragmented knowledge distributions (Chen et al., 2025; Zhang et al., 2025). Beyond sparsity, recent studies show that many layers encode overlapping transformations, and cross-layer parameter sharing (e.g., ALBERT (Lan et al., 2019)) incurs little performance loss (Ma et al., 2022; Gromov et al., 2024; Csordás et al., 2025). These observations suggest that layers capture variations of common subspaces rather than independent features. Taken together, they indicate that MoE models may conceal reusable structures obscured by routing sparsity and layer redundancy, motivating approaches that consolidate expert signals and analyze cross-layer regularities to uncover universal knowledge.

**Reusable Knowledge in Large Models.** Existing methods, such as gradient sensitivity (Molchanov et al., 2019; Sanh et al., 2020), loss-change tracking (Frankle & Carbin, 2018), and data valuation (Zhang et al., 2024; Xu et al., 2023; Wu et al., 2025a), estimate parameter importance by analyzing optimization dynamics or data influence. While these techniques provide useful signals for identifying critical weights and have been widely adopted in model compression and sparsification, they primarily reveal patterns that are either highly dispersed across parameters or aggregated at the layer level. Other approaches, including knowledge distillation (Gu et al., 2023; Xu et al., 2024) and parameter-efficient fine-tuning methods such as adapters and LoRA (Hu et al., 2022; Liu et al., 2026; Kou et al., 2025b; Li et al., 2025), improve training efficiency by reusing pre-trained weights while introducing lightweight task-specific modules. Related strategies such as selective subspace projection (Zhu et al., 2024b) or subspace-based reconstruction methods (Zhu et al., 2020) similarly focus on adapting models to specific tasks or data distributions. Overall, these

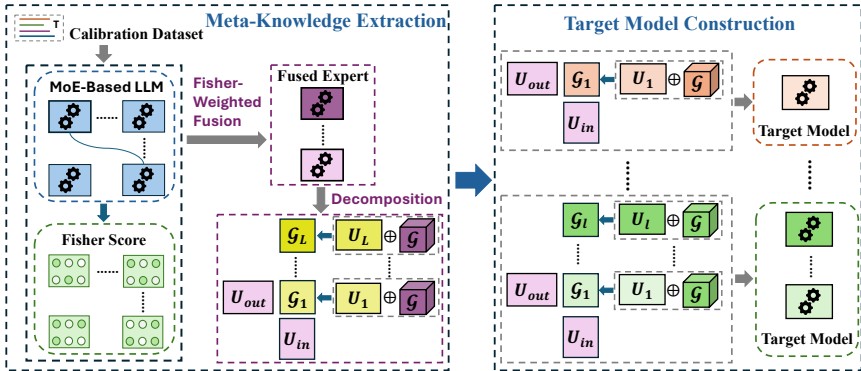

Figure 2: Overview of the *UNITE* framework. Expert signals are first consolidated within each layer via Fisher-weighted fusion, guided by calibration-based Fisher scores. The fused experts are then decomposed into shared projection matrices ($U_{in}, U_{out}$) and layer-specific cores ($\mathcal{G}_\ell$), forming universal knowledge. Finally, this knowledge is recombined with lightweight coefficients to reconstruct feed-forward modules, enabling scalable target models under different computational budgets.

lines of work emphasize task-specific adaptation and efficiency optimization rather than identifying structurally stable and reusable knowledge units that persist across architectures and tasks.

Recently, reusable paradigms such as *Learngene* (Wang et al., 2022; 2023) and *Learnware* (Zhou, 2016; Zhou & Tan, 2024) have been proposed to explicitly model transferable knowledge as modular and composable units. Learngene focuses on extracting inheritable knowledge components from an ancestry model to initialize descendant models, aiming to preserve stable functional regularities across training stages and tasks (Yuan et al., 2025; Feng et al., 2025; Guo et al., 2025). Learnware (Tan et al., 2025; Shi et al.; Tan et al., 2024), in contrast, treats pre-trained models as reusable functional assets that can be dynamically selected, composed, and deployed based on user requirements, thereby promoting model reuse at the system level. While these approaches demonstrate the promise of meta-knowledge abstraction and structured reuse, they mainly target dense architectures such as CNNs and Vision Transformers and typically operate at the layer or block granularity. Fine-grained, cross-layer, and architecture-agnostic reusable knowledge extraction for highly modular structures, such as Mixture-of-Experts models, remains underexplored.

Unlike importance scoring, compression, or distillation, *UNITE* investigates whether MoE LLMs contain *reusable, task-agnostic universal structure*. By constructing and factorizing a unified expert tensor into a shared subspace, UNITE reveals universal structure in MoEs—an aspect not addressed by prior methods.

## 3 METHOD

We introduce *UNITE* (*Universal kNowledge Integration from Task-specific Experts*), a framework that extracts and reuses universal knowledge in MoE-based LLMs. *UNITE* consists of three steps: (i) *expert importance estimation and fusion*, which consolidates fragmented expertise within each layer; (ii) *universal knowledge extraction via Tucker decomposition*, which identifies cross-layer shared structures; and (iii) *target models construction*, which validates reusability by constructing scalable target models from the extracted knowledge. Figure 2 illustrates the overall pipeline.

### 3.1 EXPERT IMPORTANCE ESTIMATION AND FUSION

In MoE-based LLMs, each layer contains multiple experts, yet only a small subset is activated for any given input, resulting in highly heterogeneous utilization across experts. A straightforward method for consolidating information within a layer is *uniform averaging*:

$$W_l^{\text{avg}} = \frac{1}{E} \sum_{i=1}^{E} W_{l,i},$$

(1)

where $W_{l,i}$ denotes the parameter matrix of expert $i$ in layer $l$. However, this method implicitly assumes equal importance across experts, which is rarely the case in practice. Empirical evidence shows that a few experts often dominate the predictions, while many others remain underutilized (Du et al., 2022; Zhang et al., 2025). Since knowledge is fragmented across experts and parameter sensitivity is not uniform, uniform averaging tends to dilute critical information and does not effectively capture the layer's expertise.

To overcome this limitation, we adapt a *Fisher information-based fusion strategy* that explicitly accounts for expert importance. The Fisher information matrix has long been employed to quantify parameter sensitivity in neural networks (LeCun et al., 1989; Martens, 2020), and here we leverage it to identify the experts whose parameters are most critical for model predictions. Formally, given a calibration dataset $\mathcal{D}$, the Fisher information of expert $i$ in layer $l$ is:

$$F_{l,i} = \mathbb{E}_{(x,y)\sim\mathcal{D}}\left[\nabla_{\theta_{l,i}} \log p(y|x;\theta) \, \nabla_{\theta_{l,i}} \log p(y|x;\theta)^\top\right], \tag{2}$$

where $\theta_{l,i}$ denotes the parameters of expert $i$ in layer $l$. Intuitively, experts with larger Fisher scores exert a stronger influence on the likelihood and thus encode more critical knowledge.

Based on these scores, we fuse experts within each layer using Fisher-weighted averaging:

$$W_l^{\mathrm{f}} = \sum_{i=1}^{E} \alpha_{l,i} W_{l,i}, \quad \alpha_{l,i} = \frac{F_{l,i}}{\sum_{j=1}^{E} F_{l,j}}. \tag{3}$$

This formulation assigns higher weights to informative experts while suppressing noise from rarely activated ones. The resulting *fused expert* provides a compact yet comprehensive summary of the layer's knowledge, offering a stronger foundation for subsequent cross-layer factorization. Importantly, the calibration dataset $\mathcal{D}$ is chosen to be *domain-general and diverse*, ensuring that the estimated importance reflects broadly transferable signals rather than overfitting to a specific downstream task.

## 3.2 Universal-Knowledge Extraction via Tucker Decomposition

Although Fisher-weighted fusion consolidates the most informative experts in each layer, substantial redundancy persists across layers, as many implement overlapping or functionally similar transformations (Gromov et al., 2024; Csordás et al., 2025). To identify a compact, transferable component, we apply Tucker decomposition to separate cross-layer shared structures from layer-specific variations. Formally, let each fused expert weight be represented as $W_l^f \in \mathbb{R}^{d_o \times d_i}$, and stack them across $L$ layers to form a third-order tensor:

$$\mathcal{W} \in \mathbb{R}^{L \times d_o \times d_i}. \tag{4}$$

We then approximate $\mathcal{W}$ via the Tucker decomposition (Kolda & Bader, 2009):

$$\mathcal{W} \approx \mathcal{G} \times_1 U_L \times_2 U_o \times_3 U_i, \tag{5}$$

where $U_L \in \mathbb{R}^{L \times r_L}$ captures layer-specific coefficients, $\mathcal{G} \in \mathbb{R}^{r_L \times r_o \times r_i}$ is the interaction tensor encoding cross-dimensional relations, and $U_o \in \mathbb{R}^{d_o \times r_o}$, $U_i \in \mathbb{R}^{d_i \times r_i}$ are projection matrices that define low-rank subspaces in the output and input dimensions.

From the Tucker decomposition, $U_o$ and $U_i$ define the common projection bases through which all layers express their transformations, while $\mathcal{G}$ with $U_L$ modulate these bases to capture layer-specific variations. To illustrate this, the weight matrix for the $l$-th layer can be reconstructed as follows:

$$W_l^f \approx U_o \, \mathcal{G}_l \, U_i^\top, \quad \mathcal{G}_l = \sum_{k=1}^{r_L} U_L[l,k] \, \mathcal{G}_{k,:,:}, \tag{6}$$

where $\mathcal{G}_l \in \mathbb{R}^{r_o \times r_i}$ denotes the slice of $\mathcal{G}$ associated with the $l$-th layer.

We therefore designate $\{U_o, U_i\}$ as the extracted *universal knowledge*, which can be interpreted as learngene, since they represent shared low-dimensional subspaces reused across layers. In contrast, $\mathcal{G}_l$ captures the layer-specific variations, enabling each layer to adapt the shared subspaces to its own functional role. This separation reflects the earlier intuition developed: MoE experts contain fragmented yet overlapping knowledge, and by consolidating them within layers and factorizing across layers, we uncover universal structures while retaining the diversity necessary for effective architectures.

### 3.3 Target Model Construction

After extracting the universal knowledge, the next question is how to operationalize this learngene component in downstream model construction. Analogous to human learning, where general principles are reused across domains, we employ the extracted universal knowledge matrices to construct target models for various downstream tasks. The central idea is straightforward: the shared matrices $U_o$ and $U_i$, obtained from the Tucker decomposition, serve as the common bases through which all layers operate, while each layer contributes its own coefficient matrix $\mathcal{G}_l$. By recombining these components, we reconstruct the feed-forward modules in a way that preserves the universal structure while allowing layer-specific adaptation.

Concretely, for the $l$-th layer of a target model, the weight matrix is reconstructed as follows:

$$\hat{W}_l = U_o \, \mathcal{G}_l \, U_i^\top, \tag{7}$$

Here, $U_o \in \mathbb{R}^{d_o \times r_o}$ and $U_i \in \mathbb{R}^{d_i \times r_i}$ are the universal knowledge matrices extracted once from the source model (i.e., the decomposed factors obtained in Eq. 5). While $\mathcal{G}_l \in \mathbb{R}^{r_o \times r_i}$ denotes the layer-specific coefficients, which are randomly initialized to provide flexibility for adaptation. After initialization, all parameters of the target models are updated jointly during *supervised fine-tuning (SFT)* on downstream datasets. This design ensures that the universal bases are reused consistently across layers, while the layer-specific components adapt to task-specific signals.

This reconstruction naturally facilitates the creation of target models with an adjustable scale. By varying the number of layers $L' \leq L$, we can build shallower models for resource-constrained environments or retain more layers for higher-capacity settings. Similarly, adjusting the Tucker ranks $(r_L, r_o, r_i)$ provides fine-grained control over parameter counts, yielding a family of models with different sizes but consistent reliance on the same universal component. In this way, *UNITE* achieves *once-for-all extraction*: universal knowledge is extracted once from a source model and subsequently reused to create target models of different scales, providing a principled way to validate its effectiveness through flexible reconstruction.

## 4 Experiment

### 4.1 Experimental Setups

**Models and Datasets.** To test whether our method can effectively extract universal knowledge from source models, we evaluate *UNITE* on three representative MoE-based LLMs that differ in scale and routing strategies: *Mixtral-8×7B* (Jiang et al., 2024), which employs 8 experts per layer with top-2 activation; *DeepSeek-MoE-16B-Base* (Liu et al., 2024), which extends to 16 experts per layer with training-aware routing for improved specialization; and *Qwen3-MoE-30B* (Yang et al., 2025), a 30B-parameter model with refined sparse expert routing to improve efficiency.

To identify universal knowledge in the source models, we compute Fisher information using the Wikitext-2 (Merity et al., 2016) as a calibration dataset, which offers general-domain coverage without biasing towards specific downstream tasks as discussed in Appendix A.4. To evaluate whether the extracted universal knowledge transfers effectively across diverse downstream tasks, we assess the constructed target models on seven widely used benchmarks: ARC-Challenge (ARC-C) and ARC-Easy (ARC-E) (Clark et al., 2018) (scientific reasoning at different difficulty levels), HellaSwag (Zellers et al., 2019) and Winogrande (Sakaguchi et al., 2021) (commonsense and logical reasoning), OpenBookQA (OBQA) (Mihaylov et al., 2018) and PIQA (Bisk et al., 2020) (scientific and physical commonsense), and GLUE-RTE (Wang et al., 2018) (natural language inference).

**Baselines.** To comprehensively evaluate our method, we compare the target models constructed by universal knowledge against two categories of baselines: small- and medium-scale pre-trained models, and model compression methods. The first category includes *T5-Base* (Raffel et al., 2020), *BERT-Large* (Devlin et al., 2019), *GPT-2* (Radford et al., 2019), and *XLM-RoBERTa* (Conneau et al., 2019), which serve as strong reference points for assessing the competitiveness of our constructed target models. The second category is represented by *Delta Decompression* (Gu et al., 2025), a recent approach that combines decomposition and pruning for effective model compression. We apply *Delta Decompression* to all source models to ensure a fair comparison across architectures.

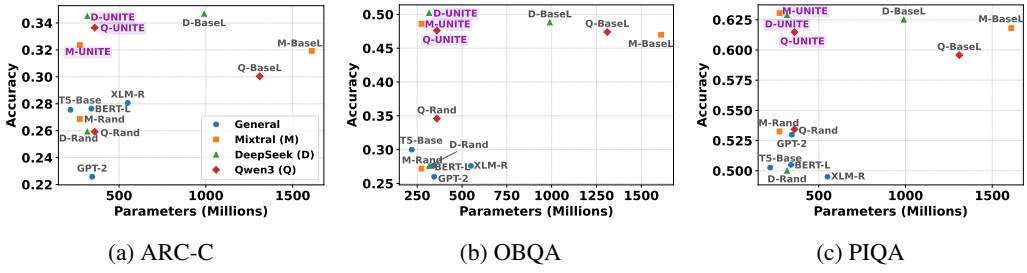

Figure 3: Comparison of *UNITE*-based target models against randomly initialized models, pre-trained baselines, and recent compression methods on downstream datasets. Here, "BaseL" denotes models obtained via *Delta Decompression* compression method.

**Implementation Details.** All the source and baseline models are loaded from the Hugging Face Model Hub [1]. We apply Tucker decomposition with a default rank of 512, and additionally experiment with ranks $\{128, 256, 512\}$ to study decomposition granularity. Target models are constructed with $\{2, 4, 6, 8, 10\}$ layers to span a range of parameter budgets. For comparison, we also include a *learning-from-scratch* baseline where the models are pretrained on 1B tokens. Both the UNITE-based target models and the baselines are subsequently fine-tuned on downstream datasets and evaluated on their validation sets using accuracy as the metric. Additional details are in Appendix A.2.

## 4.2 EVALUATING THE EFFECTIVENESS OF UNIVERSAL KNOWLEDGE

To systematically validate our method, we draw inspiration from the characteristics of human learning, such as the ability to acquire transferable principles, adapt with limited supervision, and rapidly generalize to new tasks. Accordingly, we design experiments aligned with three criteria, including *generalization*, *data efficiency*, and *convergence speed*. These evaluations directly test whether the universal knowledge extracted by *UNITE* provides a reusable foundation for efficient and robust adaptation across diverse downstream datasets.

**Generalization across Tasks.** To verify whether the universal knowledge extracted by *UNITE* enables effective transferability, we construct target models initialized with it and compare them against three categories of baselines: (1) models with the same architecture, randomly initialized followed by 1B-token pretraining and then fine-tuning, (2) small- and medium-scale pre-trained models, and (3) model compression methods on source models. Figure 3 reports representative results on downstream datasets, with the full results provided in the Appendix. Across these datasets, target models constructed with universal knowledge from *Mixtral*, *DeepSeek*, and *Qwen* cluster in the upper-left region, indicating that even with fewer parameters, UNITE-based models achieve competitive accuracy against both pretrained models and compression-based approaches.

Compared with randomly initialized models, *UNITE* yields consistent gains. On ARC-C, *Mixtral*-, *DeepSeek*-, and *Qwen*-based UNITE models improve over their random counterparts by $+5.5\%$, $+8.5\%$, and $+9.5\%$, respectively, confirming the robustness of the extracted universal knowledge. Against pretrained baselines of comparable scale, UNITE-based models achieve superior accuracy. For example, on PIQA the *Mixtral*-based UNITE attains $0.631$, surpassing BERT-Large ($0.505$) by more than $+12\%$, highlighting the efficiency of reusing universal knowledge for initialization. Compared with model compression methods, *UNITE* achieves comparable performance with substantially fewer parameters. On OBQA, the *Qwen*-based UNITE model ($360M, 0.476$) matches the accuracy of its compressed counterpart ($1.31B, 0.474$) while using only about one-fourth of the parameters. Overall, these results demonstrate that *UNITE* extracts transferable structures from source MoE models, providing a reusable initialization that generalizes across downstream datasets and delivers accuracy gains without large-scale data pre-training.

**Data Efficiency.** We next evaluate whether *UNITE* enhances data efficiency by comparing target models initialized with universal knowledge which are directly fine-tuned on downstream tasks, with randomly initialized counterparts pretrained with varying amounts of data (1B, 5B, 10B, and 20B

---

[1]https://huggingface.co/models

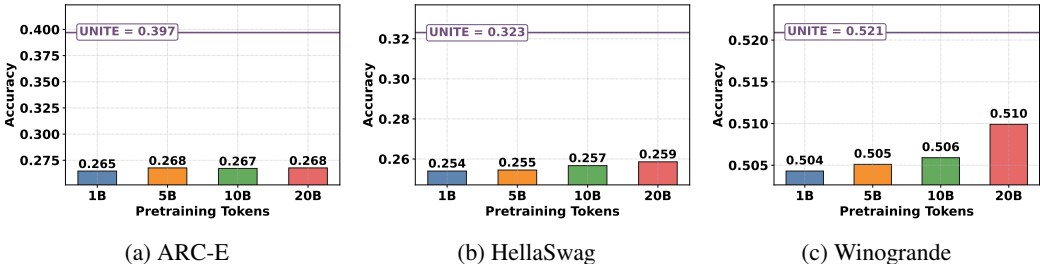

(a) ARC-E      (b) HellaSwag      (c) Winogrande

Figure 4: Data efficiency evaluation on *DeepSeek*-based target models, comparing *UNITE* with randomly initialized models pretrained on $\{1B, 5B, 10B, 20B\}$ tokens.

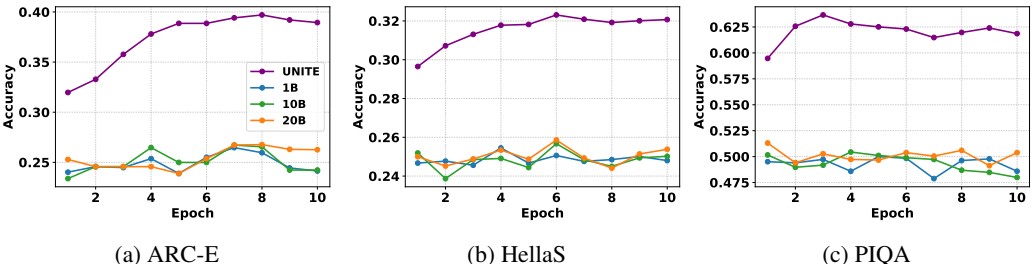

(a) ARC-E      (b) HellaS      (c) PIQA

Figure 5: Convergence speed evaluation of *DeepSeek*-based target models, comparing *UNITE* with randomly initialized models pretrained on $\{1B, 10B, 20B\}$ tokens.

tokens) before fine-tuning. As shown in Figure 4, *DeepSeek*-based target models constructed with *UNITE* consistently achieve higher accuracy than randomly initialized models, even those exposed to extensive pretraining. For example, on ARC-E, the *UNITE*-initialized model reaches $0.397$, far exceeding the best pretrained baseline ($0.268$ at 20B tokens). This shows that reusing universal knowledge provides a substantially stronger initialization, reducing reliance on costly pretraining.

Moreover, *UNITE*-based models rival or surpass small- and medium-scale pretrained models such as *T5-Base*, *BERT-Large*, *GPT-2*, and *XLM-RoBERTa*, all of which were trained on extensive corpora. This highlights the efficiency of reusing universal knowledge, which achieves competitive or superior performance without the need for extensive pretraining. Overall, these results demonstrate that *UNITE* enables data-efficient adaptation, making it particularly attractive for scenarios where large-scale pretraining is impractical.

**Convergence Speed during Fine-Tuning.** We further evaluate the convergence speed of *UNITE*-based target models by comparing them with their randomly initialised counterparts that have been pre-trained on 1B, 10B, and 20B tokens, plotting validation accuracy against training epochs. Figure 5 reports results for *DeepSeek*-based target models across downstream datasets. The curves show that *UNITE* converges substantially faster. Across all tasks, it exhibits a clear upward trajectory in the early epochs and reaches stable accuracy within 3–5 epochs. In contrast, randomly initialized models converge slowly and oscillate without significant improvement, even after 10 epochs. For instance, on ARC-E, *UNITE* surpasses $0.36$ accuracy by the fourth epoch, while the best random baseline remains below $0.27$ throughout training. These results indicate that the universal knowledge extracted by *UNITE* not only improves final accuracy but also provides a well-conditioned initialization that accelerates optimization, enabling strong performance with fewer steps and reduced training cost.

### 4.3 ABLATION STUDIES

We conduct ablation studies along two dimensions. First, we examine the core components of *UNITE*, including (i) Fisher-weighted fusion and (ii) the decomposition strategy. Second, we study scale-related factors of the target models, including (i) the number of layers and (ii) the Tucker rank.

**Fisher-weighted Fusion.** To evaluate the impact of Fisher information on expert consolidation, we compare Fisher-weighted fusion with a naive uniform averaging baseline, while keeping all other

Table 1: Ablation on a 2-layer *DeepSeek*-based target model. Top: fusion strategies. Bottom: decomposition strategies. Bold denotes the best per column.

| Method | ARC-C | ARC-E | HellaSwag | OBQA | PIQA | Winogrande | RTE |
|---|---|---|---|---|---|---|---|
| *Fusion Strategy* | | | | | | | |
| Uniform Avg | 0.3262 | 0.3378 | 0.2574 | 0.4720 | 0.5092 | 0.5098 | 0.5379 |
| *UNITE* (**Fisher**) | **0.3451** | **0.3970** | **0.3231** | **0.5020** | **0.6289** | **0.5185** | **0.5632** |
| *Decomposition Strategy* | | | | | | | |
| CP | 0.2790 | 0.2681 | 0.2577 | 0.3020 | 0.5136 | 0.5043 | 0.5271 |
| SVD | 0.2695 | 0.2668 | 0.2585 | 0.2860 | 0.5027 | 0.5075 | 0.5451 |
| *UNITE* (**Tucker**) | **0.3451** | **0.3970** | **0.3231** | **0.5020** | **0.6289** | **0.5185** | **0.5632** |

Table 2: Performance of *Qwen3*-based target models at different depths. All models share the same universal knowledge but vary in layer count. Bold marks the best result per dataset.

| Layer | Params | ARC-C | ARC-E | HellaSwag | OBQA | PIQA | Winogrande | RTE |
|---|---|---|---|---|---|---|---|---|
| 2 | 360M | 0.3365 | 0.3548 | 0.3275 | 0.476 | 0.6148 | 0.5185 | 0.5487 |
| 4 | 408M | 0.3330 | 0.4262 | 0.3592 | 0.478 | 0.6192 | **0.5280** | 0.5523 |
| 6 | 456M | 0.3313 | 0.4444 | 0.3634 | 0.490 | 0.6360 | 0.5154 | 0.5848 |
| 8 | 504M | 0.3391 | 0.4579 | **0.3645** | 0.494 | 0.6415 | 0.5193 | **0.5957** |
| 10 | 552M | **0.3494** | **0.4584** | 0.3628 | **0.506** | **0.6464** | 0.5178 | 0.5848 |

settings identical. Table 1 reports the results for a 2-layer *DeepSeek*-based target model. Fisher weighting consistently outperforms uniform averaging across all datasets. For instance, accuracy on ARC-Easy improves from 0.3378 to 0.3970, a gain of $+5.9\%$, and on PIQA it increases from 0.5092 to 0.6289, an improvement of nearly $+12\%$. These results confirm that Fisher-based fusion effectively emphasizes informative experts and suppresses redundancy, thereby enabling more reliable universal knowledge extraction.

**Decomposition Strategy.** We compare Tucker decomposition with Singular Value Decomposition (SVD) and Canonical Polyadic (CP) decomposition. As shown in Table 1, on a 2-layer *DeepSeek*-based target model, Tucker consistently achieves the best performance. For example, it improves ARC-Easy by over $+10\%$ absolute accuracy (0.3970 vs. 0.2681 with CP) and yields more than $+10\%$ gain on PIQA compared with SVD (0.6289 vs. 0.5027). These results align with Tucker's ability to capture mode-wise structure: its factor matrices recover shared input/output subspaces ($U_o$, $U_i$) while preserving layer-specific variations ($\mathcal{G}_l$). In contrast, CP oversimplifies interactions by imposing a single shared rank, and SVD cannot model multi-way correlations. Thus, Tucker provides the most suitable decomposition for isolating transferable structure from layer-level redundancy.

**Effect of Target Model Depth.** We further investigate how the scale of the constructed target models affects performance. Leveraging the flexibility of *UNITE*, we build models with depths ranging from 2 to 10 layers, where all FFN modules share the same universal knowledge matrices, but differ in their layer-specific coefficients $\mathcal{G}_l$. Table 2 reports results for *Qwen3*-based target models, with additional models provided in the Appendix.

Overall, increasing the depth leads to higher accuracy across datasets. For example, ARC-Easy improves from $0.3548$ at 2 layers to $0.4584$ at 10 layers, while PIQA increases from $0.6148$ to $0.6464$. Moderate depths (6–8 layers) already deliver strong performance, indicating a favorable balance between parameter count and accuracy. However, the improvements are task-dependent: reasoning-oriented benchmarks such as ARC-Easy benefit substantially from deeper models, whereas tasks like Winogrande and GLUE-RTE remain relatively stable across depths. This suggests that deeper models are advantageous for capturing complex reasoning structures, while shallower models suffice for tasks that rely on localized or surface-level reasoning.

Crucially, because *UNITE* incurs minimal additional cost once universal knowledge is extracted, practitioners can flexibly instantiate and evaluate models of different depths without retraining. This once-for-all property stands in contrast to distillation and compression methods, where each new scale requires costly retraining and does not always yield consistent improvements.

Table 3: Effect of Tucker rank on a *Qwen3*-based target model. Bold is the best result per column.

| Rank | ARC-C | ARC-E | HellaSwag | OBQA | PIQA | Winogrande | RTE |
|------|-------|-------|-----------|------|------|------------|-----|
| 128 | 0.3236 | 0.3860 | 0.3442 | 0.4680 | 0.6050 | 0.5107 | 0.5415 |
| 256 | 0.3262 | 0.4144 | 0.3448 | 0.4760 | 0.6175 | 0.5193 | 0.5379 |
| **512** | **0.3330** | **0.4262** | **0.3592** | **0.4780** | **0.6192** | **0.5280** | **0.5523** |

**Effect of Tucker Rank.**    We analyze the impact of Tucker rank on target model performance. As shown in Table 3 for a *Qwen3*-based model, larger ranks generally improve accuracy. For example, ARC-Easy rises from 0.386 at rank-128 to 0.4262 at rank-512 ($+4.0\%$), and RTE shows consistent improvements. However, the effect is task-dependent: on HellaSwag, the gain is marginal ($+1.5\%$), indicating diminishing returns at higher ranks. Overall, these results indicate that Tucker rank regulates the trade-off between model compactness and representational capacity. Rank-256 attains competitive accuracy with substantially fewer parameters, whereas rank-512 achieves the highest performance at a moderate increase in cost. Hence, Tucker rank functions as a tuning parameter, enabling *UNITE* to adaptively balance efficiency and accuracy across diverse deployment scenarios.

## 5    CONCLUSION

In this paper, we investigated the presence of *universal knowledge* within MoE models and introduced *UNITE* to extract and reuse it through Fisher-weighted fusion and Tucker decomposition. Extensive experiments across multiple benchmarks demonstrate that the extracted universal knowledge enables transferable initialization, improves data efficiency, accelerates convergence, and supports flexible target model construction. These results highlight *UNITE* as an effective framework for leveraging MoE models beyond pretraining and compression, providing a principled and scalable path toward building efficient downstream models.

## ACKNOWLEDGMENTS

This research was supported by the Jiangsu Science Foundation (BG2024036, BK20243012), the National Science Foundation of China (62125602, U24A20324, 92464301), the New Cornerstone Science Foundation through the XPLORER PRIZE, and the Fundamental Research Funds for the Central Universities (2242025K30024). This research was also partially supported by the Southeast University Kunpeng & Ascend Center of Cultivation and the Big Data Computing Center of Southeast University.

## ETHICS STATEMENT

This work does not involve human subjects, personal data, or sensitive user information. All datasets used are publicly available and widely adopted in prior research. Our method focuses on knowledge extraction and model efficiency in MoE architectures, without introducing risks of discrimination, bias, or privacy leakage. We have carefully followed ethical research practices and ensured compliance with data usage policies.

## REPRODUCIBILITY STATEMENT

We provide detailed descriptions of our method and experimental settings in the main text. Models, datasets, and evaluation protocols are clearly specified, and additional results are included in the appendix. To facilitate reproducibility, the key source code is included in the supplementary materials. Upon acceptance, we will release the complete codebase together with training and evaluation scripts.

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

## A  APPENDIX

### A.1  USE OF LARGE LANGUAGE MODELS (LLMS)

Large Language Models (LLMs) were used as auxiliary tools to assist in text polishing and language refinement. All ideas, methods, theoretical analysis, and experimental designs presented in this paper are solely the work of the authors.

### A.2  IMPLEMENTATION DETAILS

#### A.2.1  DATASETS

To evaluate the effectiveness and generality of our framework, we adapt widely used benchmarks spanning scientific reasoning, commonsense inference, and natural language understanding. Table 4 summarizes the datasets, including task type, dataset size, and availability of test labels. For datasets where test labels are not publicly released (e.g., PIQA, HellaSwag, Winogrande and RTE), we report results on the validation set following standard practice.

#### A.2.2  IMPLEMENTATION DETAILS FOR TARGET MODEL CONSTRUCTION

UNITE extracts universal structure only from the MoE feed-forward experts. All non-MoE components of the Transformer—including token and positional embeddings, all self-attention projections (Q/K/V/O), and LayerNorm parameters are directly copied from the pretrained source model. During downstream supervised fine-tuning, these parameters are updated jointly together with the reconstructed feed-forward modules.

#### A.2.3  RUNTIME AND MEMORY COST OF FISHER/TUCKER EXTRACTION

UNITE performs Fisher information estimation and Tucker decomposition as a **one-time, offline extraction step**. These operations are not part of training or inference and are reused for all reconstructed target models, making their cost fully amortized.

**Runtime and Memory Usage.**   We report the measured extraction overhead for the two largest MoE backbones used in our experiments:

- **DeepSeek-MoE-16B (64 experts, top-8 routing)** Fisher runtime: 1298.8 sec ($\approx$22 min), Max memory allocated: 28.1 GB (reserved: 30.9 GB).
- **Qwen3-MoE-30B (128 experts, top-2 routing)** Fisher runtime: 3355.7 sec ($\approx$56 min), Max memory allocated: 53.5 GB (reserved: 58.9 GB).

**Tucker Decomposition.**   Tucker decomposition is lightweight (under 10 minutes for all layers combined) because it operates on the fused per-layer tensor rather than the full expert parameter matrices. It is compatible with a single V100 GPU.

#### A.2.4  INFERENCE EFFICIENCY

While MoE models activate only a subset of experts during inference, their full backbones (16B–30B parameters) must still be stored and maintained in memory. In contrast, the UNITE-constructed descendant models are fully dense but substantially smaller (300M–400M parameters), resulting in significantly lower inference latency despite not relying on sparse routing.

We report measured latency for both the original MoE models and the corresponding UNITE target models:

- **DeepSeek-MoE-16B (original MoE)**: 0.539 ms/token (top-8 routing, $\sim$2.0B activated parameters).
- **UNITE-DeepSeek-L2 (317M)**: 0.020 ms/token.
- **Qwen3-MoE-30B (original MoE)**: 1.630 ms/token (top-2 routing, $\sim$0.96B activated parameters).

Table 4: Summary of datasets used in experiments. ✗indicates test labels are not publicly released, so evaluation follows the validation set.

| Dataset | Task Type | Size (Train/Dev/Test) | Test Labels |
|---|---|---|---|
| ARC-C / ARC-E | Multi-choice QA (4 options) | 2.5k / 2.5k / 1.7k | ✓ |
| HellaSwag | Sentence completion (4 options) | 39k / 10k / 10k | ✗ |
| OBQA | Multi-choice QA (4 options) | 5.0k / 0.5k / 0.5k | ✓ |
| PIQA | Physical commonsense (2 options) | 16k / 2k / 3k | ✗ |
| Winogrande | Coreference (binary) | 40k / 1.3k / 1.3k | ✗ |
| RTE | NLI (binary) | 2.5k / 0.3k / 3k | ✗ |

- **UNITE-Qwen3-L2 (360M)**: 0.022 ms/token.

These measurements show that UNITE descendant models achieve **26.7×** speedup over DeepSeek-MoE-16B and **73.7×** speedup over Qwen3-MoE-30B. The efficiency improvement comes from their dramatically reduced parameter counts rather than sparse routing, while the extracted shared subspaces preserve strong downstream accuracy.

### A.2.5 PARAMETER BREAKDOWN OF THE QWEN3-BASED 2-LAYER TARGET MODEL

For clarity, we provide the full parameter breakdown for the Qwen3-based 2-layer target model reported as 360M parameters. This model uses a hidden size of 2048 and a task-specific classifier head (instead of a tied LM head), which keeps the embedding matrix within the overall parameter budget.

**Token Embedding.** The token embedding matrix has shape $[151{,}936, 2048]$, contributing

$$151{,}936 \times 2048 = 311.16\text{M parameters.}$$

**Self-Attention (Two Layers).** Each layer includes Q/K/V/O projections and layer norms, totaling 18.88M parameters per layer:

$$2 \times 18.88\text{M} = 37.76\text{M.}$$

**MoE Feed-Forward (Two Layers).** Each layer uses shared input/output matrices plus a small Tucker core:

$$2 \times 5.11\text{M} = 10.22\text{M.}$$

**Final Normalization and Classifier.** The final LayerNorm and a $4{\times}2048$ classifier contribute approximately 0.016M parameters.

**Total.** Summing all components gives:

$$311.16\text{M} + 37.76\text{M} + 10.22\text{M} + 0.016\text{M} \approx 359.95\text{M,}$$

which agrees with the reported 360M parameter size.

### A.3 DETAILED EXPERIMENTAL RESULTS

### A.3.1 EVALUATING THE EFFECTIVENESS OF UNIVERSAL KNOWLEDGE

**Generalization across Tasks.** In the main paper, we presented representative comparisons of *UNITE* against randomly initialized baselines, pretrained models, and compression methods on a subset of benchmarks using summary plots. For completeness, we include here the extended results covering all benchmarks. Figure 6 provides additional visual comparisons, while Table 5 reports the exact numerical accuracies. The table includes results for (i) small- and medium-scale pretrained baselines, (ii) recent model compression methods, (iii) from-scratch training with pretraining on 1B tokens, and (iv) our proposed *UNITE*-based target models. Together, these results complement the main text by providing a comprehensive and reproducible view of model performance across all evaluated datasets.

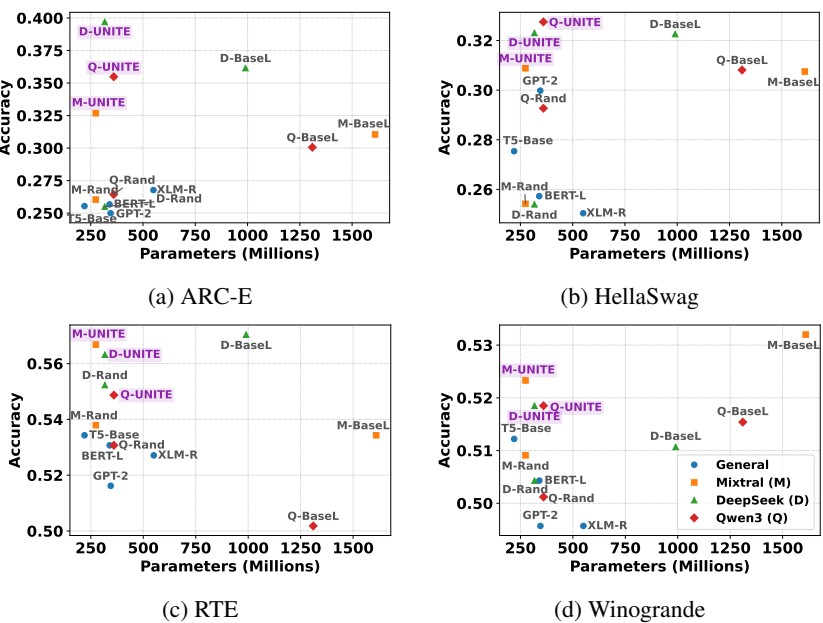

Figure 6: Extended comparison of *UNITE*-based target models against randomly initialized models, pretrained baselines, and recent compression methods on additional benchmarks. These results complement the summary plots shown in the main paper.

Table 5: Full benchmark results across all datasets. Models are grouped by source architecture (Mixtral, DeepSeek, Qwen3). "Random" denotes target models with random initialization, "Baseline" refers to pretrained or compressed models, and "Ours" corresponds to *UNITE*-based models.

| Model | Params | ARC-C | ARC-E | HellaSwag | OBQA | PIQA | Winogrande | RTE |
|---|---|---|---|---|---|---|---|---|
| **Dense Pretrained Baselines** | | | | | | | | |
| T5-Base | 220M | 0.2755 | 0.2554 | 0.2754 | 0.3000 | 0.5027 | 0.5122 | 0.5343 |
| BERT-Large | 340M | 0.2764 | 0.2567 | 0.2573 | 0.2760 | 0.5049 | 0.5043 | 0.5307 |
| GPT-2 | 345M | 0.2258 | 0.2499 | 0.2998 | 0.2600 | 0.5299 | 0.4957 | 0.5162 |
| XLM-RoBERTa | 550M | 0.2807 | 0.2677 | 0.2504 | 0.2760 | 0.4951 | 0.4950 | 0.5271 |
| **Mixtral-based Target Models** | | | | | | | | |
| Mixtral-Random | 274M | 0.2687 | 0.2604 | 0.2542 | 0.2720 | 0.5326 | 0.5091 | 0.5379 |
| Baseline (Mixtral) | 1.61B | 0.3193 | 0.3105 | 0.3075 | 0.4700 | 0.6181 | 0.5320 | 0.5343 |
| Mixtral-UNITE | 274M | 0.3236 | 0.3645 | 0.3088 | 0.4860 | 0.6306 | 0.5233 | 0.5668 |
| **DeepSeek-based Target Models** | | | | | | | | |
| DeepSeek-Random | 317M | 0.2593 | 0.2550 | 0.2540 | 0.2760 | 0.5000 | 0.5043 | 0.5523 |
| Baseline (DeepSeek) | 990M | 0.3468 | 0.3615 | 0.3226 | 0.4880 | 0.6251 | 0.5107 | 0.5704 |
| DeepSeek-UNITE | 317M | 0.3451 | 0.3970 | 0.3231 | 0.5020 | 0.6289 | 0.5185 | 0.5632 |
| **Qwen3-based Target Models** | | | | | | | | |
| Qwen3-Random | 360M | 0.2592 | 0.2643 | 0.2927 | 0.3460 | 0.5343 | 0.5012 | 0.5307 |
| Baseline (Qwen3) | 1.31B | 0.3004 | 0.3006 | 0.3081 | 0.4740 | 0.5958 | 0.5154 | 0.5018 |
| Qwen3-UNITE | 360M | 0.3365 | 0.3548 | 0.3275 | 0.4760 | 0.6148 | 0.5185 | 0.5487 |

**Data Efficiency.** In the main text, we presented representative results on a subset of datasets to demonstrate the data efficiency of *UNITE*. For completeness, we provide here the extended results covering all remaining benchmarks. Figure 7 reports accuracy on additional datasets for *DeepSeek*-based target models, as well as the full set of results for *Mixtral*-based targets. The comparison follows the same setting as in the main paper: *UNITE*-initialized models, directly fine-tuned on downstream tasks, are contrasted with randomly initialized counterparts pretrained with different amounts of data (1B, 5B, 10B, and 20B tokens). These extended results confirm the trends observed in the main text, showing that *UNITE*-based models consistently outperform their pretrained baselines while avoiding the high cost of large-scale pretraining.

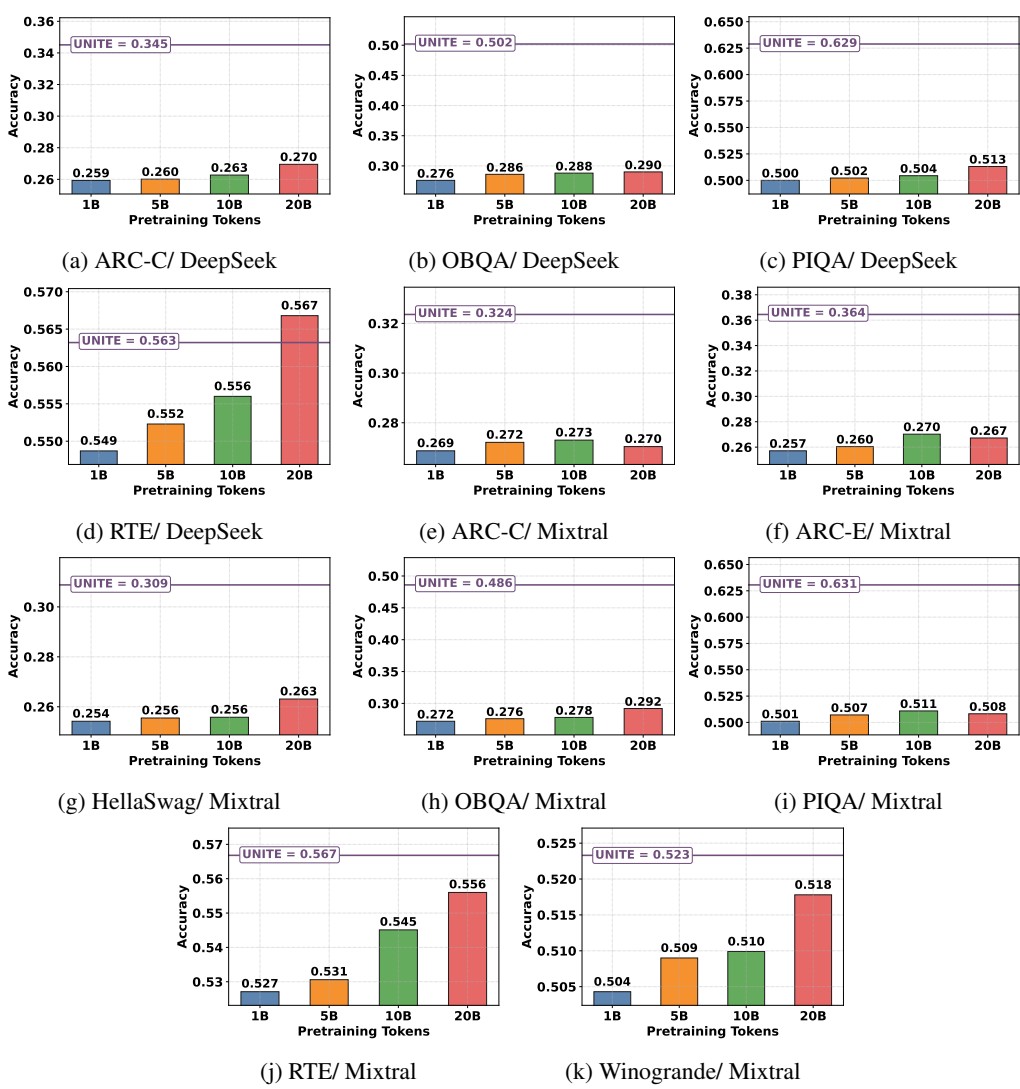

Figure 7: Extended data efficiency evaluation. Results include additional benchmarks for *DeepSeek*-based target models and the full set of benchmarks for *Mixtral*-based target models.

### A.3.2 ABLATION STUDIES

**Fisher-weighted Fusion.** In the main paper, we analyzed the role of Fisher-weighted fusion in consolidating experts using selected datasets. For completeness, we report here the extended results on 2-layer *Qwen3-* and *Mixtral*-based target models. Table 6 shows a detailed comparison between Fisher-weighted fusion and uniform averaging across all benchmarks. The results consistently demonstrate that Fisher weighting yields superior performance, confirming its effectiveness in emphasizing informative experts and reducing redundancy.

**Decomposition Strategy.** We further evaluate different decomposition strategies for consolidating universal knowledge, comparing Canonical Polyadic (CP), Singular Value Decomposition (SVD), and Tucker decomposition. Table 7 reports results for 2-layer *Qwen3-* and *Mixtral*-based target models. Across both settings, Tucker consistently achieves the best performance, yielding noticeable gains on reasoning-oriented benchmarks such as ARC-Easy and PIQA. These results indicate that Tucker is more effective at capturing multi-dimensional correlations between universal and layer-specific components, whereas CP and SVD tend to oversimplify the interactions and result in weaker transfer.

Table 6: Ablation results on Fisher-weighted fusion versus uniform averaging for 2-layer *Qwen3*- and *Mixtral*-based target models. Bold numbers indicate better performance within each pair. These results complement the main paper by providing additional benchmarks.

| Model | ARC-C | ARC-E | HellaSwag | OBQA | PIQA | Winogrande | RTE |
|---|---|---|---|---|---|---|---|
| **Qwen3-based (2-layer)** | | | | | | | |
| Uniform Avg | 0.3021 | 0.3129 | 0.3263 | 0.4740 | 0.6001 | 0.5028 | 0.5199 |
| *UNITE* (**Fisher**) | **0.3365** | **0.3548** | **0.3275** | **0.4860** | **0.6148** | **0.5185** | **0.5487** |
| **Mixtral-based (2-layer)** | | | | | | | |
| Uniform Avg | 0.3082 | 0.2500 | 0.2573 | 0.4420 | 0.6110 | 0.5146 | 0.5451 |
| *UNITE* (**Fisher**) | **0.3236** | **0.3268** | **0.3088** | **0.4860** | **0.6306** | **0.5233** | **0.5668** |

Table 7: Ablation results on decomposition strategies (CP, SVD, Tucker) for 2-layer *Qwen3*- and *Mixtral*-based target models. Tucker consistently achieves the best accuracy across benchmarks.

| Model | ARC-C | ARC-E | HellaSwag | OBQA | PIQA | Winogrande | RTE |
|---|---|---|---|---|---|---|---|
| **Qwen3-based Target Models (2-layer)** | | | | | | | |
| CP | 0.2884 | 0.3019 | 0.3238 | 0.474 | 0.6088 | 0.5075 | 0.5343 |
| SVD | 0.3090 | 0.3298 | 0.3195 | 0.470 | 0.6055 | 0.5146 | 0.5379 |
| Tucker | **0.3365** | **0.3548** | **0.3275** | **0.486** | **0.6148** | **0.5185** | **0.5487** |
| **Mixtral-based Target Models (2-layer)** | | | | | | | |
| CP | 0.2652 | 0.2677 | 0.2573 | 0.280 | 0.5049 | 0.5075 | 0.5487 |
| SVD | 0.2661 | 0.2676 | 0.2587 | 0.284 | 0.5048 | 0.5130 | 0.5415 |
| Tucker | **0.3236** | **0.3645** | **0.3088** | **0.476** | **0.6306** | **0.5233** | **0.5668** |

**Effect of Target Model Depth.** We further analyze how the depth of target models impacts performance. Table 8 reports results for *Mixtral*- and *DeepSeek*-based targets constructed with depths ranging from 2 to 10 layers. Across both architectures, increasing depth generally improves accuracy on reasoning-heavy benchmarks such as ARC-Easy and PIQA, while performance on tasks like Winogrande and RTE remains relatively stable. These findings suggest that deeper models are more effective at capturing complex reasoning structures, whereas shallower models suffice for localized reasoning tasks. Importantly, because *UNITE* enables once-for-all extraction, practitioners can flexibly instantiate models of different depths with minimal cost, in contrast to distillation or compression methods that require retraining at each scale.

**Effect of Decomposition Rank.** To evaluate the influence of decomposition granularity, we vary the rank parameter in the matrix factorization step (128, 256, and 512). Table 9 presents results for 2-layer *DeepSeek*- and *Mixtral*-based targets. Across both architectures, higher ranks generally lead to better performance, with Rank-512 consistently achieving the strongest results. Notably, on ARC-Easy, the performance of *Mixtral*-based targets improves from 0.3509 at rank-128 to 0.3645 at rank-512, while *DeepSeek*-based targets rise from 0.3302 to 0.3970 on the same benchmark. These results indicate that richer decompositions preserve more transferable information, though moderate ranks (e.g., 256) already provide competitive trade-offs between efficiency and accuracy.

### A.4 ANALYSIS ON CALIBRATION SENSITIVITY

This section provides the extended experimental results referenced in the main paper, addressing the calibration-related questions. We analyze the sensitivity of UNITE to (i) calibration data size, (ii) dataset/domain shift, (iii) approximate Fisher estimation.

### A.4.1 CALIBRATION SIZE

To study how much calibration data is required to obtain stable Fisher estimates, we vary the number of calibration sequences from 256 to 2048 using Wikitext-2. Results are reported in Table 10.

Table 8: Ablation study on the depth of target models constructed with *UNITE*. Results are reported for *Mixtral-* and *DeepSeek*-based targets with depths ranging from 2 to 10 layers. Bold indicates the best score within each block.

| Layer | Params | ARC-C | ARC-E | HellaSwag | OBQA | PIQA | Winogrande | RTE |
|---|---|---|---|---|---|---|---|---|
| **Mixtral-based Target Models** | | | | | | | | |
| 2 | 274M | 0.3236 | 0.3645 | 0.3088 | 0.486 | 0.6306 | 0.5233 | 0.5668 |
| 4 | 416M | 0.3202 | 0.3742 | 0.3113 | 0.492 | **0.6333** | **0.5406** | **0.5740** |
| 6 | 558M | 0.3142 | 0.3788 | 0.3107 | 0.486 | 0.6251 | 0.5193 | 0.5668 |
| 8 | 700M | 0.3236 | 0.3792 | 0.3109 | **0.504** | 0.6202 | 0.5257 | 0.5415 |
| 10 | 842M | **0.3305** | **0.3848** | **0.3124** | 0.492 | 0.6246 | 0.5320 | 0.5451 |
| **DeepSeek-based Target Models** | | | | | | | | |
| 2 | 317M | 0.3451 | 0.3970 | 0.3231 | 0.502 | 0.6289 | 0.5185 | 0.5632 |
| 4 | 363M | 0.3554 | 0.4101 | 0.3437 | 0.504 | 0.6572 | 0.5217 | 0.5740 |
| 6 | 409M | 0.3605 | 0.4013 | 0.3449 | 0.506 | 0.6578 | **0.5288** | **0.6101** |
| 8 | 454M | **0.3631** | 0.4110 | 0.3501 | 0.508 | 0.6703 | 0.5249 | 0.5884 |
| 10 | 500M | 0.3571 | **0.4376** | **0.3515** | **0.514** | **0.6746** | 0.5272 | 0.5957 |

Table 9: Ablation study on decomposition rank for *UNITE*-based target models. Results are reported for 2-layer *DeepSeek-* and *Mixtral*-based targets with ranks ranging from 128 to 512. Bold indicates the best score within each block.

| Model | ARC-C | ARC-E | HellaSwag | OBQA | PIQA | Winogrande | RTE |
|---|---|---|---|---|---|---|---|
| **DeepSeek (2-layer)** | | | | | | | |
| Rank-128 | 0.3356 | 0.3302 | 0.3140 | 0.466 | 0.5033 | 0.5114 | 0.5523 |
| Rank-256 | 0.3365 | 0.3455 | 0.3144 | 0.492 | 0.6213 | 0.5138 | 0.5451 |
| Rank-512 | **0.3451** | **0.3970** | **0.3231** | **0.5020** | **0.6289** | **0.5185** | **0.5632** |
| **Mixtral (2-layer)** | | | | | | | |
| Rank-128 | 0.3193 | 0.3509 | 0.3062 | 0.450 | 0.6121 | 0.5122 | 0.5487 |
| Rank-256 | 0.3219 | 0.3543 | 0.3036 | 0.478 | 0.6126 | 0.5185 | 0.5451 |
| Rank-512 | **0.3236** | **0.3645** | **0.3088** | **0.486** | **0.6306** | **0.5233** | **0.5668** |

Even very small calibration sets (around 256 sequences) provide substantial performance gains over random initialization. Increasing the calibration size brings moderate improvements but exhibits saturation beyond 1024 sequences.

**Observation.** UNITE is highly data-efficient: a few hundred sequences already produce strong universal structures. Calibration size has a sublinear effect on performance, confirming that large-scale calibration is unnecessary.

### A.4.2 CALIBRATION DATASET AND DOMAIN SHIFT

We next investigate whether the domain of the calibration dataset affects the extracted universal structures. We compare three commonly used corpora including Wikitext-2, C4, and OpenWebText, which vary substantially in cleanliness, homogeneity, and linguistic noise.

**Observation.** All datasets produce substantial improvements over random initialization. Clean and homogeneous datasets (e.g., Wikitext-2) yield the most stable Fisher estimates, while noisier datasets (C4, OpenWebText) provide slightly noisier gradients but remain effective. These results demonstrate that UNITE does not depend on dataset-specific characteristics and is robust to domain shift as long as the calibration corpus is reasonably clean.

Table 10: Effect of calibration size on downstream accuracy (DeepSeek-based 2-layer target model).

| Calibration Size | ARC-C | ARC-E | HellaSwag | OBQA | PIQA | Winogrande | Avg |
|---|---|---|---|---|---|---|---|
| Random init | 0.2593 | 0.255 | 0.254 | 0.276 | 0.5000 | 0.5043 | **0.3411** |
| wikitext-256 | 0.3185 | 0.3302 | 0.3122 | 0.480 | 0.6115 | 0.5043 | **0.4261** |
| wikitext-512 | 0.3348 | 0.3827 | 0.3183 | 0.498 | 0.6240 | 0.5140 | **0.4453** |
| wikitext-1024 | 0.3451 | 0.3970 | 0.3231 | 0.502 | 0.6289 | 0.5185 | **0.4524** |
| wikitext-2048 | 0.3468 | 0.3848 | 0.3213 | 0.500 | 0.6311 | 0.5233 | **0.4512** |

Table 11: Effect of calibration dataset on downstream accuracy (DeepSeek-based 2-layer target model).

| Dataset | ARC-C | ARC-E | HellaSwag | OBQA | PIQA | Winogrande | Avg |
|---|---|---|---|---|---|---|---|
| Random init | 0.2593 | 0.255 | 0.254 | 0.276 | 0.5000 | 0.5043 | **0.3411** |
| C4 | 0.3193 | 0.304 | 0.3132 | 0.468 | 0.6148 | 0.5012 | **0.4201** |
| OpenWebText | 0.3073 | 0.3136 | 0.3145 | 0.466 | 0.6126 | 0.4972 | **0.4185** |
| Wikitext-2 | 0.3451 | 0.3970 | 0.3231 | 0.502 | 0.6289 | 0.5185 | **0.4524** |

### A.4.3 APPROXIMATE FISHER ESTIMATION

To address concerns about the computational cost of full Fisher estimation on extremely large MoE models, we also evaluate a lightweight approximation: the diagonal Fisher, which discards all second-order parameter interactions.

**Observation.** Diagonal Fisher significantly outperforms random initialization, showing that very cheap approximations are viable. Full Fisher provides the best accuracy, reflecting the value of second-order structure, but the diagonal variant remains a practical option when computation is limited.

### A.5 COMPUTE-AWARE RANK AND DEPTH SELECTION.

The Tucker rank and the target depth are selected in two independent stages of UNITE: the rank controls the fidelity of the extracted universal structures, while the depth controls the expressive capacity of the reconstructed model. Although grid search was used in our initial experiments, our analysis reveals that both hyperparameters can be selected in a compute-aware, single-shot manner.

**Rank selection via spectral energy.** We examine the singular-value spectra of the fused MoE FFN tensors. Both DeepSeek-MoE-16B and Qwen3-MoE-30B exhibit slow spectral decay, and the retained spectral energy increases monotonically with the Tucker rank. This trend matches the accuracy improvements observed in Table 3. The spectral-energy profiles in Figure 8a therefore offer a compute-aware heuristic: *select the largest Tucker rank permitted by GPU memory*, eliminating the need for grid search.

**Depth selection via diminishing returns.** For the Qwen3-based target models, we evaluate depths from 2 to 20 layers. Figure 8b shows a clear diminishing-returns pattern: accuracy increases rapidly from 2 to 8 layers, plateaus around 8–10 layers, and yields only marginal gains thereafter. This behaviour mirrors the "knee point" commonly observed in scaling-law analyses of language models.

Taken together with the spectral-energy analysis, these results show that both Tucker rank and target depth can be selected without grid search—rank via spectral-energy heuristics and depth via scaling-law style diminishing-return curves.

### A.6 THEORETICAL DERIVATION

We now provide a theoretical justification for the generalization advantage of *UNITE*. The analysis focuses on two aspects: (i) Fisher-weighted fusion yields more stable estimators of ideal

Table 12: Comparison of diagonal Fisher and full Fisher.

| Method | ARC-C | ARC-E | HellaSwag | OBQA | PIQA | Winogrande | Avg |
|--------|-------|-------|-----------|------|------|------------|-----|
| Diagonal Fisher | 0.3322 | 0.3264 | 0.3152 | 0.494 | 0.6121 | 0.4972 | **0.4295** |
| Full Fisher | 0.3451 | 0.3970 | 0.3231 | 0.502 | 0.6289 | 0.5185 | **0.4524** |

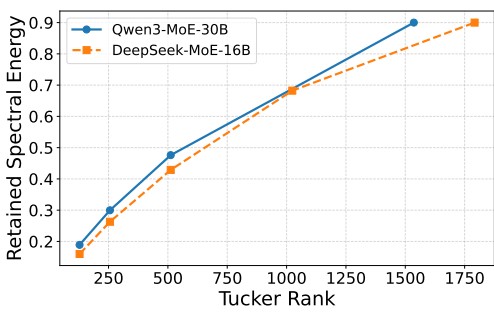

(a) Spectral energy retained vs. Tucker rank.

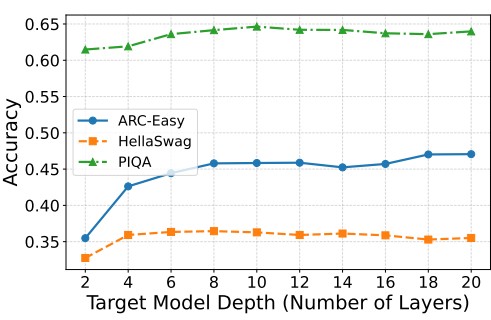

(b) Accuracy vs. target model depth based.

Figure 8: (a) Spectral–energy curves of the fused MoE FFN tensor, illustrating the monotonic increase in retained energy as the Tucker rank grows, which provides a compute-aware heuristic for selecting the rank; (b) downstream accuracy of Qwen3-based UNITE models across different depths, showing a clear diminishing-return pattern and a scaling-law–style knee point.

layer transformations compared to uniform averaging, and (ii) decomposition with shared bases and lightweight layer-specific coefficients reduces hypothesis class complexity, leading to tighter generalization bounds.

**Theorem A.1 (Generalization Advantage of UNITE's Knowledge Extraction)** *Let $\mathcal{H}_{MoE}$ denote the hypothesis class of a source Mixture-of-Experts (MoE) model with $L$ layers and $E$ experts per layer, with parameters $\{W_{l,i}\}_{l=1,i=1}^{L,E}$. Let $\mathcal{D}$ be the underlying data distribution, and suppose each expert is an unbiased but noisy estimate of an ideal transformation $W_l^*$ for its layer:*

$$W_{l,i} = W_l^* + \mathcal{E}_{l,i}, \quad \mathbb{E}[\mathcal{E}_{l,i}] = 0, \ \text{Var}(\text{vec}(\mathcal{E}_{l,i})) = \Sigma_{l,i}.$$

*The UNITE framework constructs target models from a hypothesis space $\mathcal{H}_{UNITE}$ by (i) Fisher-weighted fusion of experts and (ii) Tucker decomposition of the fused matrices into universal bases $\{U_o, U_i\}$ with layer-specific coefficients $\{G_l\}$. Then:*

1. *(**Stability of Fisher-weighted Fusion**) Assume $\text{Tr}(\Sigma_{l,i}) \propto 1/F_{l,i}$, where $F_{l,i}$ is the Fisher information of expert $i$. Then the Fisher-weighted fusion*

$$W_l^f = \sum_{i=1}^{E} \alpha_{l,i} W_{l,i}, \quad \alpha_{l,i} = \frac{F_{l,i}}{\sum_j F_{l,j}},$$

*is the minimum-variance unbiased estimator of $W_l^*$ among all convex combinations. In particular,*

$$\mathbb{E}\|W_l^f - W_l^*\|_F^2 \ \leq \ \mathbb{E}\|W_l^{\text{avg}} - W_l^*\|_F^2,$$

*where $W_l^{\text{avg}} = \frac{1}{E} \sum_{i=1}^{E} W_{l,i}$ is the uniform average.*

2. *(**Complexity Reduction via Decomposition**) Given $\{W_l^f\}_{l=1}^{L}$, Tucker decomposition extracts universal bases $U_o \in \mathbb{R}^{d_o \times r_o}$, $U_i \in \mathbb{R}^{d_i \times r_i}$ shared across layers, while each layer $l$ only learns a small coefficient matrix $G_l \in \mathbb{R}^{r_o \times r_i}$:*

$$W_l' = U_o G_l U_i^\top.$$

*Define $\mathcal{H}_{UNITE} = \{h(x; \{G_l\})\}$ and $\mathcal{H}_{Naive} = \{h(x; \{W_l\})\}$. Since the number of learnable parameters in $\{G_l\}$ is significantly smaller than in $\{W_l\}$ while $U_o, U_i$ remain fixed, the empirical Rademacher complexity satisfies*

$$\mathcal{R}_n(\mathcal{H}_{UNITE}) \ll \mathcal{R}_n(\mathcal{H}_{Naive}),$$

*leading to a tighter generalization bound.*

PART 1: STABILITY OF FISHER-WEIGHTED FUSION

We formally analyze why Fisher-weighted fusion provides a more stable estimate of the latent transformation compared to uniform averaging.

Let each expert's parameter matrix in layer $l$ be denoted as $W_{l,i} \in \mathbb{R}^{d_o \times d_i}$, modeled as a noisy observation of an ideal, latent transformation $W_l^*$:

$$W_{l,i} = W_l^* + \mathcal{E}_{l,i},$$

where $\mathcal{E}_{l,i}$ is a zero-mean noise matrix with covariance $\Sigma_{l,i}$, i.e.,

$$\mathbb{E}[\mathcal{E}_{l,i}] = 0, \quad \mathrm{Var}(\mathrm{vec}(\mathcal{E}_{l,i})) = \Sigma_{l,i}.$$

We further assume independence across experts, i.e.,

$$\mathrm{Cov}(\mathcal{E}_{l,i}, \mathcal{E}_{l,j}) = 0 \quad \text{for } i \neq j.$$

For any weighted estimator $\tilde{W}_l = \sum_{i=1}^{E} \beta_i W_{l,i}$ with $\sum_i \beta_i = 1$, the expected squared Frobenius error relative to $W_l^*$ is:

$$\mathbb{E}\big[\|\tilde{W}_l - W_l^*\|_F^2\big] = \mathbb{E}\left[\left\|\sum_{i=1}^{E} \beta_i \mathcal{E}_{l,i}\right\|_F^2\right]$$

$$= \sum_{i=1}^{E} \beta_i^2 \, \mathbb{E}[\|\mathcal{E}_{l,i}\|_F^2]$$

$$= \sum_{i=1}^{E} \beta_i^2 \, \mathrm{Tr}(\Sigma_{l,i}).$$

To connect this to the UNITE framework, we posit that the Fisher information $F_{l,i}$ of an expert reflects its stability with respect to the underlying data distribution. Intuitively, a highly informative expert has lower variance. This is consistent with the Cramér–Rao bound, which establishes that Fisher information is inversely related to the variance of an unbiased estimator:

$$\mathrm{Tr}(\Sigma_{l,i}) \propto \frac{1}{F_{l,i}}.$$

Under this assumption, the optimal weights that minimize the expected estimation error are given by

$$\beta_{l,i}^* \propto \frac{1}{\mathrm{Tr}(\Sigma_{l,i})} \propto F_{l,i}.$$

Normalizing the weights yields

$$\alpha_{l,i} = \frac{F_{l,i}}{\sum_{j=1}^{E} F_{l,j}},$$

which are exactly the weights employed in Fisher-weighted fusion.

By the property of the minimum-variance unbiased estimator (MVUE), any weighting proportional to the inverse variance achieves the lowest possible variance among linear unbiased estimators. Consequently, the Fisher-weighted estimator satisfies

$$\mathbb{E}\big[\|W_l^f - W_l^*\|_F^2\big] \leq \mathbb{E}\big[\|W_l^{\mathrm{avg}} - W_l^*\|_F^2\big],$$

where $W_l^f$ is the Fisher-weighted fused matrix and $W_l^{\mathrm{avg}}$ is the uniform average.

Thus, Fisher-weighted fusion provides a statistically more stable estimate of the ideal transformation by up-weighting reliable experts and suppressing noisy ones, forming the foundation for robust universal knowledge extraction in *UNITE*.

PART 2: COMPLEXITY REDUCTION FROM TUCKER DECOMPOSITION

After obtaining a stable set of fused transformations $\{W_l^f\}_{l=1}^L$, the UNITE framework applies Tucker decomposition to uncover universal low-dimensional bases shared across layers. Concretely, each fused matrix $W_l^f \in \mathbb{R}^{d_o \times d_i}$ is approximated as

$$W_l^f \approx U_o G_l U_i^\top,$$

where $U_o \in \mathbb{R}^{d_o \times r_o}$ and $U_i \in \mathbb{R}^{d_i \times r_i}$ are universal orthogonal bases shared across layers, and $G_l \in \mathbb{R}^{r_o \times r_i}$ is the layer-specific core. In constructing a target model of depth $L'$, only the set of compact cores $\{G_l\}_{l=1}^{L'}$ are trainable, while the bases $\{U_o, U_i\}$ remain fixed.

Formally, the hypothesis space of UNITE can be expressed as

$$\mathcal{H}_{\text{UNITE}} = \left\{ h(x; \{G_l\}) \,\middle|\, G_l \in \mathbb{R}^{r_o \times r_i}, \, l = 1, \ldots, L' \right\},$$

with $r_o \ll d_o$ and $r_i \ll d_i$. In contrast, a naive fine-tuning approach without decomposition directly optimizes the full fused matrices $\{W_l^f\}$, leading to the hypothesis class

$$\mathcal{H}_{\text{Naive}} = \left\{ h(x; \{W_l\}) \,\middle|\, W_l \in \mathbb{R}^{d_o \times d_i}, \, l = 1, \ldots, L' \right\}.$$

To analyze the generalization implications, we compare their Rademacher complexities. Since the Rademacher complexity $\mathcal{R}_n(\mathcal{H})$ grows with the square root of the number of trainable parameters, we obtain:

$$\mathcal{R}_n(\mathcal{H}_{\text{UNITE}}) \propto \sqrt{L' \cdot r_o \cdot r_i}, \quad \mathcal{R}_n(\mathcal{H}_{\text{Naive}}) \propto \sqrt{L' \cdot d_o \cdot d_i}.$$

Because Tucker decomposition enforces $r_o \ll d_o$ and $r_i \ll d_i$, the hypothesis class induced by UNITE is substantially less complex:

$$\mathcal{R}_n(\mathcal{H}_{\text{UNITE}}) \ll \mathcal{R}_n(\mathcal{H}_{\text{Naive}}).$$

By standard results in statistical learning theory, for any hypothesis $h \in \mathcal{H}$ and with probability at least $1 - \delta$, the true risk satisfies

$$R(h) \leq \hat{R}_n(h) + 2\mathcal{R}_n(\mathcal{H}) + O\left( \sqrt{\frac{\log(1/\delta)}{n}} \right),$$

where $\hat{R}_n(h)$ denotes the empirical risk. A smaller hypothesis class complexity thus yields a tighter generalization bound.

In summary, Tucker decomposition in UNITE restricts learning to compact layer-specific cores while reusing universal bases, thereby lowering model complexity. This reduction ensures that models fine-tuned with limited data are less prone to overfitting and generalize more reliably compared to naive full-matrix tuning.

