# OpenReview forum: "UNITE: Universal kNowledge Integration from Task-specific Experts"
_ICLR.cc/2026/Conference — ICLR 2026 Poster_

### Official Review · Reviewer_UrjS · 2025-10-26

**Soundness:** 3
**Presentation:** 2
**Contribution:** 3
**Rating:** 6
**Confidence:** 3

**Summary:**

The paper proposes UNITE, a framework that fuses MoE experts via Fisher-weighted aggregation and applies Tucker decomposition to extract shared low-rank subspaces that encode universal knowledge, from which it reconstructs a dense model training only small per-layer cores.

**Strengths:**

The paper is clearly written and well-organized, with fluent language that make the presentation easy to follow. The contribution sounds novel.

**Weaknesses:**

The method introduces Fisher estimation and Tucker decomposition. What is the efficiency of these operations? The authors should provide time-efficiency comparisons against baselines.

Does the method depend on the calibration dataset? Can it remain robust under different calibration datasets?

MoE models enjoy the advantage of activating only a small fraction of parameters at inference. After integration, how does the inference time of the resulting model compare to the original—does it lose this advantage?

Seems that the method can be viewed as a form of model distillation from an MoE teacher to a dense student. Could the authors include experimental comparisons against standard distillation baselines?

**Questions:**

The authors claim that universal knowledge is captured by the matrices U_o, U_i. What metrics or evidence substantiate this claim?

---

> ### Author Response · Authors · 2025-11-21
> **Response to Reviewer UrjS (1/2)**
>
> We thank the reviewer for the detailed feedback. We provide responses to each point below and will reflect all clarifications in the revised manuscript.
>
>
> **W 1. Efficiency of Fisher Estimation and Tucker Decomposition**
>
> Below we report the measured wall-clock time and memory usage used in our study. Both Fisher information estimation and Tucker decomposition are **one-time, offline extraction** steps and do not contribute to the training or inference cost of the target models.
>
> **Fisher information estimation**
> - DeepSeek-MoE-16B: **1298.8 sec (~22 min), max memory 28.1 GB**
> - Qwen3-MoE-30B: **3355.7 sec (~56 min), max memory 53.5 GB**
>
> **Tucker decomposition**
> - **<10 minutes** total for all layers (on a single V100 GPU)
>
> After extraction, the universal structures can be reused to initialize target models of any depth without recomputing Fisher or Tucker, making the overhead fully amortized across all target models.
>
> To contextualize efficiency, standard pretrained baselines require substantially larger compute budgets:
> - BERT-Large (340M) requires 4 days of training on 16 Cloud TPUs during pretraining.
> - T5-Base (220M) is pretrained on approximately 1T tokens using large-scale TPU clusters, with multi-week training typical for this setup.
>
> These baselines incur hundreds to thousands of GPU/TPU-hours before fine-tuning. In contrast, UNITE requires **around one hour of offline computation** to extract universal structures from 16B–30B MoE models, after which downstream fine-tuning proceeds in exactly the same way as for all baselines.
>
> ---
>
> **W 2. Whether UNITE Depends on the Calibration Dataset**
>
> Fisher information is inherently data dependent, so we examine whether the choice of calibration dataset affects the extracted universal structures. We evaluate UNITE with three corpora, namely C4, OpenWebText, and Wikitext-2, while keeping all other steps identical. The downstream results are:
>
> | Calibration Data | ARC-C | ARC-E | HellaS. | OBQA | PIQA | Winog. | Avg |
> |------------------|-------|-------|-----------|------|-------|------------|-------|
> | Random   | 0.2593 | 0.2550 | 0.2540 | 0.2760 | 0.5000 | 0.5043 | 0.3414 |
> | C4            | 0.3193 | 0.3040 | 0.3132 | 0.4680 | 0.6148 | 0.5012 | 0.4201 |
> | OpenWebText  | 0.3236 | 0.3345 | 0.3158 | 0.4800 | 0.6126 | 0.5083 | 0.4291 |
> | Wikitext-2  | 0.3451 | 0.3970 | 0.3231 | 0.5020 | 0.6289 | 0.5185 | 0.4524 |
>
> These datasets differ substantially in linguistic stability:
> - **Wikitext-2** is clean and well-structured, yielding stable gradients with low variance.
> - **OpenWebText** contains moderate noise and mixed writing styles, resulting in slightly higher variance.
> - **C4** is highly heterogeneous and noisy, which is expected to make Fisher estimation less stable.
>
> Consistent with these properties, Wikitext-2 gives the strongest results, OpenWebText is slightly lower, and C4 performs the weakest. Crucially, UNITE consistently improves over random initialization across all three datasets, showing that it does not rely on any specific corpus. In practice, a dataset that provides reasonably stable gradients is sufficient for reliable Fisher estimation.
>
> ---
>
> **W 3. Whether UNITE Loses the Inference-Time Sparsity Advantage of MoE Models**
>
> UNITE extracts shared subspaces from a pretrained MoE model and uses them to construct smaller dense target models. These target models serve two purposes: they verify the transferability of the extracted universal structure and provide compact models that can be deployed under lower compute budgets.
>
> Although MoE models activate only a small fraction of parameters during inference, the full backbone still needs to be stored and maintained in memory. For 16B–30B MoE models, this storage footprint is significantly larger than that of the 300M–400M UNITE models. As a result, even without sparse routing, the constructed dense models achieve substantially lower inference latency. Measured latency for the original MoE backbones and the corresponding UNITE models is as follows:
>
> **DeepSeek-MoE-16B (original MoE)**
> - Latency per token: 0.539 ms
> - Activated parameters (top-8): ~2.0B
> - Full model size: 16B parameters
>
> **UNITE target model (DeepSeek-L2, 317M)**
> - Latency per token: 0.020 ms
>
> **Qwen3-MoE-30B (original MoE)**
> - Latency per token: 1.630 ms
> - Activated parameters (top-2): ~0.96B
> - Full model size: 30B parameters
>
> **UNITE target model (Qwen3-L2, 360M)**
> - Latency per token: 0.022 ms
>
> These results show that, despite being fully dense, the UNITE models deliver large inference-time gains due to their dramatically smaller parameter counts. Concretely, they provide:
>
> - 26.7× speedup over DeepSeek-MoE-16B
> - 73.7× speedup over Qwen3-MoE-30B
>
> This efficiency improvement comes from reduced model size rather than sparse routing, while the extracted universal structures preserve strong downstream accuracy.

---

> ### Author Response · Authors · 2025-11-21
> **Response to Reviewer UrjS (2/2)**
>
> **W 4. Comparison with Standard Distillation Baselines**
>
> We clarify that UNITE is not a distillation method. UNITE is designed to investigate whether universal structures can be extracted from a MoE model and then reused to construct target models of different sizes. This requires only a one-time, task-agnostic extraction using Fisher fusion and Tucker decomposition. In contrast, knowledge distillation relies on task-specific supervised training and must be repeated separately for every desired student architecture. More details are in our response to Reviewer WC1f (W 1).
>
> To directly address the reviewer’s request, we implement a standard distillation baseline using the same teacher (DeepSeek-MoE-16B) and the same student architecture (a 2-layer dense model). The student’s parameters are obtained:
>
> - **Distill:** the student is randomly initialized and trained using KL + MSE distillation from the teacher.
> - **UNITE:** the student’s FFN layers are initialized from the extracted universal structures.
>
> All other training setting are the same to ensure a controlled comparison. The results are shown below:
>
> | Model  | ARC-C | ARC-E | HellaS. | OBQA | PIQA   | Winog. | Avg |
> |----|---|---|---|---|----|------|----|
> | Distill  | 0.2670  | 0.2706   | 0.2589 | 0.2900 | 0.5239 | 0.5275 | 0.3563  |
> | UNITE | **0.3451**    | **0.3970** | **0.3231** | **0.5020** | **0.6289** | **0.5185** | **0.4524** |
>
> UNITE surpasses standard distillation across all tasks, indicating that the extracted universal structures provides a significantly more effective initialization than distillation. These results confirm that UNITE complements, rather than replicates, the role of distillation and offers an efficient, reusable mechanism for transferring cross-expert and cross-layer structure from MoE models.
>
> ---
>
> **Q 1. What Evidence Supports That $(U_o, U_i)$ Capture Universal Knowledge**
>
> We acknowledge that there is no recognised quantitative metric for “universal knowledge”. Precisely because such a metric is missing, our work aims to narrow this gap by providing an empirical and systematic investigation. Below, we first justify why universal knowledge is expected to exist in LLMs, and then explain why the matrices $U_o$ and $U_i$ capture such knowledge.
>
> Prior analyses suggest that Transformer layers exhibit substantial cross-layer similarity, indicating the presence of recurring structure across depth. For example, [1] report that sharing parameters across all layers results in only minor degradation. Mode-connectivity studies further show that independently trained Transformer checkpoints can be connected through smooth, low-loss paths within a common representation space [2]. In addition, investigations of layer efficiency find that many FFN layers can be compressed or even bypassed with limited performance impact [3].
>
> Motivated by this, we examine whether MoE FFN layers contain extractable shared structure. To investigate this, we first aggregate the information that is dispersed across different experts within each layer. After consolidation, we obtain a fused tensor: $W \in \mathbb{R}^{L \times d_o \times d_i}$, whose three modes capture variation across layers, outputs, and inputs. To extract the shared structures, we apply Tucker decomposition, a multilinear generalization of PCA, which factorizes $W$ into mode-wise components. Prior work [4] shows that the principal components of such multi-way tensors correspond to shared structures across layers. In this context, the factor matrices $U_o$ and $U_i$ recover the dominant input/output directions that appear consistently across layers, making them natural candidates for layer-invariant universal structures.
>
> We further validate this interpretation empirically. Rank ablation studies show that increasing the Tucker rank, and thus retaining more shared directions, leads to improvements in downstream tasks. Using the *same* $(U_o, U_i)$, target models with different depths (2/4/8 layers) generalize robustly across six benchmarks. Furthermore, UNITE-initialized models require fewer supervised samples and converge faster than randomly initialized models. These observations together indicate that the initialization encodes informative shared structures.
>
> While there is no formal metric for “universal knowledge”, the combination of (i) prior evidence of shared structures in Transformers, (ii) the optimality properties of Tucker as a multilinear PCA, and (iii) our empirical results across rank ablation, generalization, data efficiency, and convergence collectively support the interpretation that $U_o$ and $U_i$ capture the dominant, reusable computational structures of MoE-based LLMs.
>
> [1] A Lite BERT for Self-Supervised Learning of Language Representations.
> [2] Generalized Linear Mode Connectivity for Transformers.
> [3] Do Language Models Use Their Depth Efficiently?
> [4] Traditional and Heavy-Tailed Self-Regularization in Neural Networks.

---

> > ### Comment · Reviewer_UrjS · 2025-11-27
> >
> > We appreciate the authors’ response and will maintain the positive score.

---

> > > ### Author Response · Authors · 2025-11-27
> > >
> > > We appreciate your follow-up evaluation and the continued positive assessment. Your earlier feedback on the efficiency of the extraction process, the role of calibration data, and the relation to distillation-based approaches helped us refine the presentation in the revision. We thank you for the thoughtful comments and the time you invested in reviewing our submission.

---

### Official Review · Reviewer_QWnw · 2025-10-29

**Soundness:** 2
**Presentation:** 2
**Contribution:** 2
**Rating:** 4
**Confidence:** 3

**Summary:**

The paper addresses the challenge of fragmented and redundant expertise within Mixture-of-Experts (MoE) Large Language Models (LLMs). It argues that, similar to human learning, MoE models likely encode transferable, task-agnostic knowledge (universal knowledge) that is obscured by the layer-wise redundancy and expert sparsity. UNITE systematically extracts and reuses universal knowledge from MoE-based LLMs. The method involves: Intra-layer Consolidation (with Fisher-weighted fusion )and Cross-layer Factorization(with Tucker decomposition). Experiments on three MoE-based LLMs (Mixtral, DeepSeek-MoE, Qwen3-MoE) demonstrate that UNITE-based models consistently outperform randomly initialized baselines.

**Strengths:**

1. The recognized issues of fragmented and redundant knowledge in MoE models is an interesting topic.
2. The paper thoroughly evaluates UNITE across three MoE models (Mixtral, DeepSeek, Qwen3), multiple benchmarks (ARC, HellaSwag, PIQA, etc.), and three key criteria: generalization, data efficiency, and convergence speed. Ablation studies validate each component (Fisher fusion, Tucker decomposition, rank, depth).

**Weaknesses:**

1. The motivation is not clear to me. I did not understand what "overlap" is between LLM
2.  The definition is not clear. For example, what is the definition of knowledge?

3. "By analogy, MoE-based LLMs may also exhibit similar properties: although each expert is designed to specialize, their parameters are not entirely independent and may contain overlaps, redundancies, and recurring transformations that implicitly capture transferable patterns." I am not sure the experts are specialized in moe-structure.

4. The computational cost of Fisher information calculation and Tucker decomposition on large MoE models is not discussed. This could be a practical barrier for very large models, and the paper does not analyze the trade-off between extraction cost and downstream benefits.

**Questions:**

## questions

1. What is the computational and memory overhead of the Fisher information calculation and Tucker decomposition steps, especially for larger MoE models (e.g., Qwen3-MoE-30B)? Is the extraction process scalable to models with hundreds of experts?

2. What are the advantages compared to the Tucker decomposition of the CP decomposition?

---

> ### Author Response · Authors · 2025-11-21
> **Response to Reviewer QWnw (1/2)**
>
> We thank the reviewer for the thoughtful comments. We address each point below and will reflect all clarifications in the revised manuscript.
>
>
> **W 1 & 3. Clarifying the Motivation, the Notion of “Overlap”, and Whether MoE Experts Are Specialized**
>
> Although the *“overlap”* is not explicitly adopted in most MoE papers, multiple empirical studies report phenomena that closely align with what we describe as *expert-level* and *layer-level* overlap.
>
> At the expert level, although MoE designs aim for specialization, multiple studies show that experts often behave similarly rather than forming fully independent functional modules. Many experts exhibit highly similar activation patterns [1], hierarchical clustering reveals that subsets can be merged with minimal degradation [2], and routing imbalance frequently causes under-trained experts to converge toward correlated behaviors [3]. These results indicate that experts are not always specialized in practice and can be substantially redundant.
>
> At the layer level, it has been shown that Transformers contain cross-layer redundancy. [4] shows that sharing parameters across all layers leads to minimal performance loss, suggesting that FFN layers repeatedly reuse similar low-dimensional subspaces. Furthermore, [5] shows that adjacent layers often perform highly similar transformations and can be compressed or merged with negligible impact on accuracy, providing additional evidence of redundancy at the layer level.
>
> In this paper, we use *“overlap”* to refer to these empirically observed forms of recurring structures:
> (i) **Expert-level overlap**, where experts within the same MoE layer behave similarly rather than forming fully independent functional modules; and
> (ii) **Layer-level overlap**, where FFN layers across depth learn recurring patterns and operate within shared low-dimensional subspaces.
>
> This motivates the investigation into whether such recurring structures can be explicitly extracted and reused as universal knowledge. UNITE applies Fisher-weighted consolidation to reduce fragmented expert and Tucker decomposition to identify shared cross-layer structures. The resulting shared structures transfer consistently across downstream tasks, indicating that they capture reusable structures rather than incidental redundancy.
>
> [1] Exploring Expert Specialization through Unsupervised Training in Sparse Mixture of Experts.
> [2] Retraining-Free Merging of Sparse Mixture-of-Experts via Hierarchical Clustering.
> [3] Advancing Mixture-of-Experts Efficiency and Scale.
> [4] A Lite BERT for Self-Supervised Learning of Language Representations.
> [5] Do Language Models Use Their Depth Efficiently?
>
> ---
>
> **W 2. Clarification on the Meaning of “Knowledge”**
>
> In this work, “knowledge” is used in an operational sense. It denotes the reusable computational structure that a pretrained MoE-based LLM develops through large-scale pretraining. Although the model is trained on diverse linguistic and reasoning data, many of the resulting internal patterns—such as abstraction, compositional regularities, and common feature transformation behaviors—emerge repeatedly across experts and across layers.
>
> Within UNITE, the extracted “knowledge” corresponds to the components that (i) consistently generalize across tasks, (ii) remain stable across depth, and (iii) can be isolated and reused as transferable inductive capacity. In other words, it refers to the shared structural patterns that support efficient adaptation to new tasks, rather than task-specific outputs or semantic information.

---

> ### Author Response · Authors · 2025-11-21
> **Response to Reviewer QWnw (2/2)**
>
> **W 4. Computational Cost and Downstream Trade-offs**
>
> In UNITE, both Fisher information estimation and Tucker decomposition are **one-time, offline extraction steps** steps. They do not appear in training or inference, and their cost is amortized across all target models regardless of scale. In our experiments, Fisher extraction was computed using two NVIDIA H100 GPUs, and Tucker decomposition was on a single V100 GPU:
>
> - DeepSeek-MoE-16B: **1298.8 seconds (~22 minutes)**
> - Qwen3-MoE-30B: **3355.7 seconds (~56 minutes)**
> - Tucker decomposition: **<10 minutes total**
>
> Fisher extraction is also feasible on older hardware such as V100 GPUs, with identical memory requirements and proportionally longer runtime. In terms of the trade-off between extraction cost and downstream benefit, this roughly one-hour offline computation enables initialization of target models of different sizes, making the amortized per-model cost negligible.
>
> For context, standard pretraining budgets for the 200M–500M dense baselines in our evaluation are substantially higher:
>
> - BERT-Large (340M) requires 4 days of training on 16 Cloud TPUs during pretraining.
> - T5-Base (220M) is pretrained on approximately 1T tokens using large-scale TPU clusters, with multi-week training typical for this setup.
>
> These baselines require hundreds to thousands of GPU/TPU-hours before fine-tuning. In contrast, UNITE’s extraction cost is modest, hardware-efficient, incurred only once, and downstream fine-tuning follows the same procedure as all baselines, ensuring a fair comparison.
>
> ---
>
> **Q 1. The Computational and Memory Cost, and Scalability to Larger MoE Models**
>
> The runtime and memory requirements for Fisher extraction on the two MoE backbones used in our experiments are:
>
> - **DeepSeek-MoE-16B (64 experts, top-8 routing)**
>   Fisher extraction time: **1298.8 sec (≈22 min)**
>   Max memory allocated: **28.1 GB** (reserved: **30.9 GB**)
>
> - **Qwen3-MoE-30B (128 experts, top-2 routing)**
>   Fisher extraction time: **3355.7 sec (≈56 min)**
>   Max memory allocated: **53.5 GB** (reserved: **58.9 GB**)
>
> Tucker decomposition adds <10 minutes in total because it operates on the fused per-layer tensor rather than the full FFN matrices.
>
> Regarding scalability, UNITE has already been applied to a model with 128 experts (Qwen3-MoE-30B). In practice, scalability is limited primarily by available GPU memory—Fisher extraction computes per-parameter gradients for each active expert but does not introduce algorithmic bottlenecks that prevent scaling to larger expert counts. The same procedure extends to models with more experts whenever memory capacity allows. Overall, the extraction overhead is modest and incurred only once, while its results can be reused across all reconstructed target models and all downstream tasks.
>
> ---
>
> **Q 2.Tucker Decomposition vs CP**
>
> Our choice of Tucker decomposition is directly motivated by the objectives of UNITE. After consolidating experts within each MoE layer, we aim to determine whether FFN layers contain universal structures that can be represented through shared input/output subspaces. To investigate this, we require a decomposition method that separates shared projection directions from layer-specific variations in a principled manner.
>
> The fused MoE tensor  $W \in \mathbb{R}^{L \times d_o \times d_i}$  contains three semantically distinct modes (layer, output, input). Tucker decomposition naturally aligns with this structure by assigning a factor matrix to each mode $(G_L, U_o, U_i)$. This enables UNITE to:
> (i) the factor matrices $U_o$ and $U_i$ characterize the dominant output and input subspaces that are consistently expressed across layers, and therefore serve as candidates for universal, layer-invariant structure; while
> (ii) $G_L$ encodes the layer-specific variation, capturing how each layer combines these shared subspaces.
> This mode-wise decomposition allows UNITE to isolate the common computational structures of MoE FFN layers in a mathematically principled manner, while still retaining the flexibility to model fine-grained differences across depth.
>
> In contrast, CP decomposition imposes **a single shared rank across all modes** and expresses the tensor as a sum of rank-1 components that jointly couple layer, input, and output dimensions. This entanglement prevents clean separation of mode-specific subspaces and does not reflect the functional organization of MoE FFNs. CP is also known to be numerically unstable in high-dimensional settings, whereas Tucker provides well-conditioned decompositions with controllable multilinear ranks [1,2].
>
> The empirical comparison in Table 1 reflects this structural distinction: Tucker consistently achieves higher downstream accuracy than CP under identical rank settings, indicating that it better preserves the shared structures relevant to UNITE.
>
> [1] Tensor Decompositions and Applications.
> [2] Tensor Decomposition for Signal Processing and Machine Learning.

---

### Official Review · Reviewer_WC1f · 2025-10-31

**Soundness:** 2
**Presentation:** 2
**Contribution:** 2
**Rating:** 4
**Confidence:** 4

**Summary:**

This paper proposes UNITE, a framework that extracts universal knowledge from MoE-based LLMs and uses it to initialize smaller dense models for downstream tasks. The method applies Fisher-weighted fusion to consolidate experts within each layer, then uses Tucker decomposition to extract shared projection matrices and layer-specific cores across layers. Further experiment results shows the effectiveness of the proposed method.

**Strengths:**

1. Extracting universal knowledge from MoE models is a compelling research direction.
2. The experiments effectively demonstrate the method's effectiveness across several MoE architectures and benchmarks.

**Weaknesses:**

1. **Limited technical novelty**: Both Fisher information weighting and Tucker decomposition are well-established techniques. The main contribution is their combination for MoE models, which feels incremental. The paper does not sufficiently differentiate itself from prior work on model compression, knowledge distillation.
2. **Critical architectural details missing**: The paper focuses on MoE feed-forward layers but omits key implementation details. How are embedding and attention layers initialized in target models? When constructing a model with L'=2 layers from a source with L=32 layers, which layers are selected and how?
3. **Unfair baseline comparisons**: The paper compares UNITE models (initialized from state-of-the-art MoE models like DeepSeek-16B and Qwen3-30B) against outdated baselines such as T5-Base, BERT-Large, and GPT-2. Since the source MoE models benefit from years of architectural improvements and much larger, higher-quality pretraining data, the performance gains may reflect source model quality rather than effective knowledge extraction. Fairer comparisons would include recent similarly-sized models.

**Questions:**

Another point is how the Qwen3-based 2-layer model has only 360M parameters when the embedding layer alone should exceed this? Please provide a complete parameter breakdown,including embeddings, attention, and MLP components.

---

> ### Author Response · Authors · 2025-11-21
> **Response to Reviewer WC1f (1/2)**
>
> We thank the reviewer for the helpful comments. We provide point-by-point responses and will incorporate them in revision.
>
> **W 1. Technical Novelty**
>
> To explain why we employ Fisher information weighting and Tucker decomposition, we first describe the key motivation for our study. Recent empirical studies on MoE architectures indicate that MoE layers exhibit two structural properties: (i) **intra-layer fragmentation**, where many experts behave redundantly or remain under-specialized [1,2,3]; and (ii) **cross-layer redundancy**, where FFN blocks across different layers display highly overlapping transformations [4,5]. However, most of these studies are diagnostic, suggesting that MoE models may contain deeper shared information—potentially an extractable, task-agnostic universal structures, instead of attempting to explicitly explore, extract or reuse shared knowledge.
>
> To narrow such gap, we propose UNITE to  **systematically explore, extract, and validate universal knowledge within MoE-based LLMs,** which is the key contribution and novelty of our study. To investigate this possibility, UNITE decomposes the problem into two stages aligned with these intra-layer and cross-layer phenomena. Although many alternative techniques could in principle be used, our methodological choices are driven by the need to extract the universal structure, where Fisher information weighting and Tucker decomposition effectively address these requirements.
>
> - **Identifying which experts meaningfully contribute within each layer.**
>    Given that experts within a layer are fragmented, a consolidation mechanism is required. Fisher-weighted fusion provides this by aggregating expert parameters **according to their importance**, rather than assuming uniform contribution.
>
> - **Extracting the shared structures across layers from the fused experts.**
>    Since layers exhibit overlapping or functionally similar transformations, a method is needed to recover the shared structures. Tucker decomposition achieves this by factorizing the fused tensor into **shared input/output subspaces** and **layer-specific variations**, providing exactly the mechanism required to extract universal structures.
>
> After consolidating experts and extracting the shared structures, we further investigate how this universal structures/knowledge can be *used*. We employ the extracted universal structures to construct target models of different depths, enabling deployment under varying downstream compute budgets. When evaluated across a wide range of downstream tasks, these target models exhibit consistent performance across tasks and model scales, demonstrating the **generalizability and effectiveness** of the shared structures/knowledge.
>
> To sum up, while Fisher weighting and Tucker decomposition are established tools, the contribution of this work lies in the unified **consolidate–extract–use** procedure we introduce for uncovering and validating universal structures in MoE-based LLMs, rather than in the individual techniques themselves.
>
> This perspective differs from *compression* and *distillation*. Compression removes redundant parameters to preserve a full-function model, and distillation transfers most of the teacher model’s behavior via substantial training. Neither approach targets the identification of universal structures/knowledge that can be generating **flexible, small-scale target models** tailored to downstream compute budgets—an ability that UNITE provides by design. This exploration also offers insights that may benefit future work in model compression and distillation.
>
> [1] Exploring Expert Specialization through Unsupervised Training in Sparse Mixture of Experts.
> [2] Retraining-Free Merging of Sparse Mixture-of-Experts via Hierarchical Clustering.
> [3] Advancing Mixture-of-Experts Efficiency and Scale.
> [4] A Lite BERT for Self-Supervised Learning of Language Representations.
> [5] Do Language Models Use Their Depth Efficiently?

---

> ### Author Response · Authors · 2025-11-21
> **Response to Reviewer WC1f (2/2)**
>
> **W 2. Missing Architectural Details**
>
> The details are given below and the code is in the supplementary materials.
>
> (1) Initialization of embedding and attention layers.
>
> UNITE extracts shared structure only from the MoE FFNs. All other components—token and positional embeddings, self-attention modules, and LayerNorm parameters—are directly copied from the pretrained model and jointly updated with the reconstructed FFNs during downstream training.
>
> (2) Constructing a target model with $L' < L$.
> When building a small target model (e.g., $L' = 2$ from a 32-layer MoE), no source-layer indices are selected. For the $l$-th layer of the target model, we reuse the shared structures $(U_o, U_i)$ extracted once together with a layer-specific matrix $\mathcal{G}_l \in \mathbb{R}^{r_o \times r_i}$ to construct the FFN:  $\hat{W}_l = U_o\, G_l\, U_i^{\top}.$
>
> This design keeps $(U_o, U_i)$ shared across all layers while allowing each layer to learn its own knowledge through $\mathcal{G}_l$, and the target depth $L'$ does not correspond to any subset of the original $L$ layers (lines 254–262).
>
> ---
>
> **W 3. Baseline Fairness**
>
> Classical dense baselines are included to show the gap between UNITE’s lightweight reconstructed models and widely used pretrained dense models of similar scale. We also evaluate compression baselines applied to the same MoE backbone as UNITE. As Figure 3 shows, UNITE fall in the upper-left region of the accuracy–parameter trade-off, providing competitive accuracy with significantly fewer parameters.
>
> To strengthen fairness, we add two recent dense models of comparable size: MiniLLM-340M and Qwen1.5-500M. MiniLLM/DeepSeek-MoE-16B belong to the same period, and Qwen1.5/Qwen3 share data pipelines and design. UNITE is competitive with them, confirming that the gains come from effective extraction of shared structure rather than backbone advantages.
>
> | Model              | Params | ARC-C  | ARC-E  | HellaS. | OBQA  | PIQA  | Winog. | Avg     |
> |-------------------|--------|--------|--------|-----------|-------|-------|------------|---------|
> | MiniLLM-340M    | 340M   | 0.2652 | 0.2647 | 0.2596 | 0.3180| 0.5212| 0.4957 | 0.3541  |
> | UNITE-DeepSeek | 317M | 0.3451 | 0.3970 | 0.3231 | 0.5020| 0.6289| 0.5185 | 0.4524  |
> | Qwen1.5-0.5B     | 500M   | 0.2721 | 0.2930 | 0.2638 | 0.4040| 0.5310| 0.5162 | 0.3800  |
> | UNITE-Qwen3    | 503M | 0.3391 | 0.4579 | 0.3645 | 0.4940| 0.6415| 0.5193  | 0.4694  |
>
> ---
>
> **Q 1. The parameter count of the Qwen3-based 2-layer model**
>
> We appreciate the request for clarification. The Qwen3-based 2-layer model uses a hidden size of 2048 and a task-specific classifier head rather than a tied LM head, which keeps the embedding parameters within budget. A full breakdown is given below.
>
> - Token embedding:
>
>   Embed_tokens.weight of shape [151,936, 2048], contributing151,936 × 2048 = 311.16M parameters.
> - Self-attention (2 layers):
>
>   Each layer’s Q/K/V/O projections and normalization sum to 18.88M, yielding 37.76M total.
> - MoE FFN (2 layers):
>
>   Each reconstructed FFN uses shared-in/out matrices plus a small core tensor, totaling 5.11M per layer or 10.22M total.
> - Final norm + classifier:
>   A final LayerNorm and a 4×2048 classifier add ≈0.016M parameters.
>
> Summing these components: 311.16M + 37.76M + 10.22M + 0.016M ≈ 359.95M, which matches the reported 360M. We will include this breakdown in the revision for clarity.

---

> > ### Comment · Reviewer_WC1f · 2025-11-27
> >
> > Thanks for your response. My concerns have been resolved, and I've decided to raise my score to 6.

---

> > > ### Author Response · Authors · 2025-11-27
> > >
> > > We appreciate your careful reevaluation and your decision to raise the score. Your earlier comments regarding the architectural specification, baseline configuration, and parameter accounting were valuable in improving the precision and clarity of the revised manuscript. We are grateful for the time and attention you devoted to reviewing our work.

---

### Official Review · Reviewer_rKwF · 2025-11-01

**Soundness:** 3
**Presentation:** 3
**Contribution:** 2
**Rating:** 6
**Confidence:** 3

**Summary:**

### Briefly summarization of **UNITE**

UNITE introduces a novel framework for extracting and reusing universal knowledge from Mixture-of-Experts (MoE) based large language models. The authors observe that while MoE architectures enable sparse activation and scalability, they also lead to fragmented expertise and redundancy across layers. Inspired by how humans generalize domain-specific experiences into transferable reasoning skills, UNITE aims to uncover shared, task-agnostic structures embedded within expert parameters.

### Main Contributions of **UNITE**

- **Universal Knowledge Extraction**
  First framework to *systematically extract* **task-agnostic, reusable knowledge** from MoE-based LLMs by combining **Fisher-weighted fusion** and **Tucker decomposition**.

- **Efficient Downstream Adaptation**
  Enables **once-for-all** extraction → build **lightweight target models** of *any depth* without retraining, achieving **strong performance with fewer parameters**.

- **Superior Performance**
  Outperforms **pretrained baselines** (e.g., BERT, GPT-2) and **compression methods** on diverse tasks (science, commonsense, NLI) with **+6–12% accuracy gains**.

- **Improved Training Efficiency**
  Boosts **data efficiency** (no large pretraining needed) and **convergence speed** (fewer epochs to reach peak performance).

- **Theoretical Justification**
  Provides **generalization bounds** showing Fisher-weighted fusion reduces variance and Tucker decomposition lowers model complexity.

**Strengths:**

1. Novel & Timely Idea
First to frame “universal knowledge” inside sparse MoE transformers and give a concrete, tensor-decomposition recipe to extract it—directly relevant as trillion-parameter MoEs proliferate.

2. Solid Empirical Sweep
Tests three different MoE back-bones (Mixtral, DeepSeek, Qwen3) on seven varied benchmarks; ablates every major design choice (fusion weighting, decomposition rank, target depth); consistently beats strong pretrained and compression baselines while using ¼–⅓ the parameters.

3. Theoretical Backing
Supplies generalization theorems that justify Fisher-weighted fusion as minimum-variance unbiased estimator and Tucker compression as Rademacher-complexity reducer—rare for a systems-oriented NLP paper.

**Weaknesses:**

1. Downstream Tasks Skew toward QA/CLS
Evaluation focuses on multiple-choice and sentence-pair classification; no generation, long-context, or multilingual benchmarks, leaving extrapolation to other modalities unclear.

2. Fisher Computation Overhead
Requires a full forward–backward pass on a calibration dataset to estimate expert importance; could be expensive for very large MoEs and is sensitive to the choice of that dataset (only WikiText-2 is tested).

3. Rank & Depth Chosen Empirically
Tucker ranks and target depths are grid-searched on validation sets; no principled way to predict optimal compression given a compute budget, which limits “off-the-shelf” adoption.

**Questions:**

1. Generation & long-context evaluation
All reported downstream tasks are either multiple-choice QA or sentence-pair classification. Could you provide results on generative objectives (e.g., summarization, open-ended QA) or long-context benchmarks to verify that the extracted universal bases remain useful when the output space is unconstrained or the input length grows beyond the calibration window?

2. Calibration cost & data sensitivity
The Fisher-information estimates require a full forward–backward pass over a calibration corpus. For trillion-parameter MoEs this can be prohibitive, and only WikiText-2 has been tested. How does the quality of the extracted bases vary with (a) calibration size, (b) domain shift, and (c) cheaper approximations such as diagonal Fisher or gradient-free importance metrics? Is there a “minimal sufficient” calibration set or an online variant that amortizes the cost?

3. Rank/depth selection without grid search
Tucker ranks and target depth are currently chosen by grid search on validation accuracy. Is there a theoretically grounded or compute-budget-aware procedure (e.g., eigen-gap heuristic, MDL principle, or a single-shot sensitivity analysis) that predicts the optimal compression ratio before any downstream fine-tuning, so that practitioners can deploy UNITE in an “off-the-shelf” fashion?

---

> ### Author Response · Authors · 2025-11-21
> **Response to Reviewer rKwF (1/2)**
>
> We thank the reviewer for the helpful comments. We respond to each point below and will integrate the corresponding clarifications into the revision.
>
> **W/Q 1. Additional Evaluation on Generative Benchmarks**
>
> To further evaluate whether the extracted universal bases remain useful in generative settings, we tested three instruction-oriented datasets, which involve summarization, reasoning, and instruction following. The results are shown below:
>
> | Initialization | Dolly | SelfInst | Vicuna |
> |--------------|-------|----------|--------|
> | Random         | 10.56 | 5.49     | 8.04   |
> | UNITE          | 14.81 | 7.83     | 11.83  |
>
> Across three benchmarks, UNITE outperforms random initialization, indicating that the extracted shared structures are not limited to classification settings and can generalize to more open-ended generation tasks.
>
> ---
>
> **W/Q 2. Calibration Cost & Data Ssensitivity**
>
> To assess UNITE’s sensitivity to the calibration datasets, we perform ablations along the three axes suggested by the reviewer: (a) calibration **data size**, (b) calibration **dataset/domain**, and (c) **cheaper Fisher approximations**. All other steps remain unchanged.
>
>  **(a) Calibration Data Size**
>
> We used four Wikitext-2 subsets (**256 / 512 / 1024 / 2048** sequences). Downstream performance (DeepSeek-L2) is:
>
> | Calibration size | ARC-C | ARC-E | HellaS. | OBQA | PIQA | Winog. | Avg |
> |---------------|--------|--------|-----------|-------|--------|-------------|-------|
> | Random | 0.2593 | 0.2550  | 0.2540 | 0.2760 | 0.5000 | 0.5043 | **0.3411** |
> | wikitext-256 | 0.3185 | 0.3302 | 0.3122 | 0.484 | 0.6115 | 0.5083  | **0.4275** |
> | wikitext-512 | 0.3348 | 0.3827 | 0.3183 | 0.498 | 0.6240  | 0.514 | **0.4453** |
> | wikitext-1024 | 0.3451 | 0.3970  | 0.3231| 0.502 | 0.6289 | 0.5185 | **0.4524** |
> | wikitext-2048 | 0.3468 | 0.3848 | 0.3213 | 0.500 | 0.6311 | 0.5233 | **0.4512** |
>
> Performance improves with more calibration data, but even small corpora (e.g., 256 sequences) already yield substantial gains over random initialization. This shows that UNITE does not require a large calibration set, a few hundred sequences are sufficient.
>
> ---
>
> **(b) Calibration Dataset / Domain Shift**
>
> We compared **Wikitext-2** (clean, homogeneous), **OpenWebText** (moderate noise), and **C4** (heterogeneous and noisy). The downstream results are:
>
> | Calibration Data | ARC-C | ARC-E | HellaS. | OBQA | PIQA | Winog. | Avg |
> |--------------|-------|-------|-----------|------|-------|------------|-------|
> | Random | 0.2593 | 0.2550 | 0.2540 | 0.2760 | 0.5000 | 0.5043 | 0.3414 |
> | C4  | 0.3193 | 0.3040 | 0.3132 | 0.4680 | 0.6148 | 0.5012 | 0.4201 |
> | OpenWebText  | 0.3236 | 0.3345 | 0.3158 | 0.4800 | 0.6126 | 0.5083 | 0.4291 |
> | Wikitext-2 | 0.3451 | 0.3970 | 0.3231 | 0.5020 | 0.6289 | 0.5185 | 0.4524 |
>
> Wikitext-2 provides the best results due to its clean style, but both C4 and OpenWebText still can extract stable universal structures and remain clearly above random initialization. This indicates that UNITE does not depend on a specific corpus and is robust under domain shift as long as the calibration set provides reasonably stable gradients.
>
> ---
>
> **(c) Cheaper Fisher Approximations and Practical Cost Considerations**
>
> In practice, the computational overhead of UNITE is modest. The calibration step consisting of Fisher estimation followed by Tucker decomposition—is a one-time offline procedure performed once per source model. After extraction, the shared structures are reused to construct target models of any depth, so the cost is amortized across all target models. Measured runtime for the full extraction pipeline is as follows:
> - **DeepSeek-MoE-16B**: 22 minutes
> - **Qwen3-MoE-30B**: 56 minutes
> - **Tucker decomposition**: <10 minutes
>
> Compared with standard LLM pretraining, which typically requires hundreds to thousands of GPU-hours even for 200M–500M dense models, this one-time extraction cost is negligible.
>
> Following the reviewer’s suggestion, we also tested a lightweight approximation—diagonal Fisher, which drops all second-order interactions:
>
> | Method | ARC-C | ARC-E  | HellaS. | OBQA  | PIQA  | Winog. | Avg |
> |------------------|--------|---------|-----------|-------|--------|-------------|----------------|
> | Diagonal Fisher  | 0.3322 | 0.3264  | 0.3152    | 0.494 | 0.6121 | 0.4972      | **0.4295**     |
> | Full Fisher  | 0.3451 | 0.3970  | 0.3231    | 0.502 | 0.6289 | 0.5185      | **0.4524**     |
>
>  Full Fisher performs best, indicating that second-order structure carries useful signal. Diagonal Fisher remains well above random initialization, showing that inexpensive approximations are viable. Together, these results demonstrate that UNITE’s calibration procedure is both practical and robust.
>
> ---
>
> Across all three axes—calibration size, dataset/domain, and Fisher approximation—UNITE remains consistently robust. We will include these ablations and a discussion of calibration dataset requirements in the revised manuscript.

---

> ### Author Response · Authors · 2025-11-21
> **Response to Reviewer rKwF (2/2)**
>
> **W/Q 3. Rank/Depth Selection**
>
> Our current use of grid search reflects the exploratory of this work—specifically, examining how rank and depth affect the transferability of the extracted universal structures. We agree that a more principled, compute-aware selection rule would further improve the practical applicability of UNITE.
>
> UNITE proceeds in two independent stages. The Tucker **rank** is chosen during the extraction stage and controls the fidelity with which universal structures are recovered from the fused MoE tensor. The **depth** is chosen in the reconstruction stage, where varying the number of layers allows us to evaluate the expressiveness and transferability of the extracted universal structures. Thus, rank affects extraction quality, while depth affects reconstruction capacity.
>
> ---
>
> **Towards principled rank selection.**
>
> The reviewer’s suggestion aligns with our observations. Our eigen-spectrum analysis of the fused MoE tensors shows a slow decay of singular values. Retaining 90\% of the spectral energy requires a rank around 1.5k–1.7k for DeepSeek and Qwen3. At practical rank ranges, the retained energy is:
>
> - **DeepSeek:**
>   - r=128 → 0.1599, r=256 → 0.2630, r=512 → 0.4289
> - **Qwen3:**
>   - r=128 → 0.1891, r=256 → 0.2997, r=512 → 0.4761
>
> These values match the trend observed in Table 3, where larger ranks consistently yield higher accuracy. This suggests a simple, compute-aware heuristic: practitioners may select the largest Tucker rank allowed by memory and compute budgets, without performing grid search.
>
> ---
>
> **Towards principled depth selection.**
>
> A similar pattern appears when varying target-model depth. The downstream accuracy (Qwen3, 2–20 layers) follows a clear diminishing-returns curve: accuracy improves rapidly from 2→8 layers, flattens around 8–10, and exhibits little gain thereafter. This mirrors the “knee point’’ behavior commonly reported in scaling-law analyses of LLM depth.
>
> | Task         | d2     | d4     | d6     | d8     | d10    | d12    | d14    | d16    | d18    | d20    |
> |--------------|--------|--------|--------|--------|--------|--------|--------|--------|--------|--------|
> | ARC-Easy     | 0.3548 | 0.4262 | 0.4444 | 0.4579 | 0.4584 | 0.4588 | 0.4524 | 0.4571 | 0.4702 | 0.4706 |
> | HellaSwag    | 0.3275 | 0.3592 | 0.3634 | 0.3645 | 0.3628 | 0.3592 | 0.3612 | 0.3587 | 0.3529 | 0.3551 |
> | PIQA         | 0.6148 | 0.6192 | 0.6360 | 0.6415 | 0.6464 | 0.6420 | 0.6418 | 0.6372 | 0.6360 | 0.6398 |
> | Winogrande   | 0.5185 | 0.5280 | 0.5154 | 0.5193 | 0.5178 | 0.5185 | 0.5138 | 0.5107 | 0.5257 | 0.5130 |
>
>
> Taken together with our spectral analysis for rank selection, these results suggest that both Tucker rank and target depth can be chosen without grid search by using spectral-energy heuristics for determining the rank and scaling-law–style diminishing-return curves for determining the depth.

---

### Comment · Area_Chair_v7td · 2025-11-26
**Reviewer & Author Discussion**

Dear Reviewers,

We kindly encourage you to **review and respond to the authors’ rebuttals**. Your timely feedback is important for ensuring a fair and thorough review process. Thank you for your contributions to ICLR 2026.

Thank you very much for your time and support.

Best regards,


 Area Chair v7td

---

### Author Response · Authors · 2025-11-30
**Summary of Review Status Prior to Score Reset**

Dear Area Chair,

I hope you are doing well. We appreciate your handling of the submission during the recent system issue, and we also thank the reviewers for their constructive evaluations. To support an informed continuation of the review process, we summarize below the review status prior to the score reset.

- The initial reviews converged on several themes. **Reviewer rKwF** queried the breadth of downstream evaluation and the principled choice of Tucker rank and target-model depth. **Reviewers QWnw and WC1f** requested clearer motivation and sharper definitions of *“overlap”* and *“knowledge”*. **Reviewers WC1f and UrjS** emphasized the need for fair and complete baseline comparisons. Additional questions concerned computational aspects, particularly calibration-data requirements and the efficiency of Fisher estimation and Tucker decomposition (**rKwF, QWnw, UrjS**).

- In our rebuttal, we addressed these points through **new generative-task experiments**, **expanded calibration-robustness analyses**, **calibration-data ablations**, and **spectral-energy–based justification** for Tucker rank and depth selection *(see W 1/2/3 to Reviewer rKwF)*. We refined the concept of universal structure by **clarifying the motivation**, **tightening definitions of “overlap’’ and “knowledge’’** *(see W 1 to Reviewer WC1f; W 1/3 to Reviewer QWnw)*, and **providing structural interpretations of the target model** *(see W 2 to Reviewer WC1f)*. Baseline completeness was strengthened through **additional comparisons**, including controlled distillation baselines *(see W 4 to Reviewer UrjS)* and **new pretrained models of comparable size** *(see W 3 to Reviewer WC1f)*. We also added detailed **efficiency analyses** for Fisher estimation, Tucker decomposition, and inference-time*(see W 4 / Q 1 to Reviewer QWnw; W 3 to Reviewer UrjS)*.

- These updates establish **UNITE as a principled framework that extracts universal knowledge from MoE-based LLMs and uses this structure to construct target models of varying sizes**, which show consistent performance across downstream tasks. The **consolidate–extract–reuse pipeline** further enables a single extraction to support multiple model configurations, highlighting the generality and practical value of the identified shared structure.

- Before the discussion, **two reviewers (rKwF and UrjS)** had assigned a score of **6**. Prior to the leakage incident, **Reviewer WC1f** also increased the score to **6**, noting that all initial concerns had been fully resolved. Reviewer UrjS likewise maintained a positive evaluation after the rebuttal. Although **Reviewer QWnw** did not submit a follow-up comment, all points raised were addressed comprehensively in our response, and several overlapped with issues already acknowledged and accepted by the other reviewers. Accordingly, the concerns across the four reviewers have been incorporated into the revised manuscript.

Thank you again for your time and consideration. We also appreciate the reviewers’ constructive feedback, which has meaningfully strengthened the manuscript. We remain happy to provide any additional clarification that may assist the decision process.

Best regards,
Authors of Submission 10133.

---

### Meta-Review · Area_Chair_Dw5y · 2025-12-16

**Summary:**

The paper aims to address the challenges of fragmented and redundant expertise within MoE in LLM. The authors propose a method that consists of 1) intra-layer consolidation with Fisher-weighted fusion and 2) cross-layer factorization with Tucker decomposition.

Strengths identified by reviewers include, its novelty, comprehensive experiments, compelling research direction, and clear writing.

The reviewers raise several concerns and the major concern is that the Fisher weights computation and Tucker decomposition can lead to high computational cost. Additionally, reviewers also have concerns on the evaluations on generation benchmarks or compared with more recent methods and robustness with different calibration datasets. Initially, two reviewers assign positive scores with two reviewers suggested borderline reject. After rebuttal, one reviewer confirmed maintaining positive score and one reviewer confirmed raising the score to borderline accept.

In overall, as most of initial review scores are positive and most of concerns have been addressed, the AC recommended acceptance of the paper.

**Reviewer Concerns:**

The authors provided very detailed feedback and additional experimental results during the rebuttal. Most of concerns have been addressed, including 1) lack of evaluation on generation, long-context, or multilingual benchmarks; 2) potentially high computational cost of fisher computation; 3) Rank/depth selection without grid search; 4) Limited technical novelty (as it is built on both well-established techniques Fisher information weighting and Tucker decomposition); 5) Critical architectural details missing; 6) Unfair baseline comparisons; 7) Dependence on calibration dataset and the robustness across datasets; 8) MoE's efficiency changes; 9) lack of comparisons to standard distillation baseline.

**Reviewer Scores:**

Initially, the reviewer scores are mixed (two borderline accepts and two borderline rejects). Reviewer UrjS has confirmed maintaining positive score and reviewer WC1f confirmed raising the score to 6. Other reviewers would have raised or maintained their scores.

---

### Decision · Program_Chairs · 2026-01-26

Accept (Poster)